# Single-molecule observation of ATP-independent SSB displacement by RecO in *Deinococcus radiodurans*

**Jihee Hwang[1], Jae-Yeol Kim[2], Cheolhee Kim[3], Soojin Park[1], Sungmin Joo[4], Seong Keun Kim[1]\*, Nam Ki Lee[1]\***

[1]Department of Chemistry, Seoul National University, Seoul, Republic of Korea; [2]Laboratory of Chemical Physics, National Institute of Diabetes and Digestive and Kidney Diseases, National Institutes of Health (NIH), Bethesda, United States; [3]Daegu National Science Museum, Daegu, Republic of Korea; [4]Department of Physics, Pohang University of Science and Technology, Pohang, Republic of Korea

**Abstract** *Deinococcus radiodurans* (DR) survives in the presence of hundreds of double-stranded DNA (dsDNA) breaks by efficiently repairing such breaks. RecO, a protein that is essential for the extreme radioresistance of DR, is one of the major recombination mediator proteins in the RecA-loading process in the RecFOR pathway. However, how RecO participates in the RecA-loading process is still unclear. In this work, we investigated the function of drRecO using single-molecule techniques. We found that drRecO competes with the ssDNA-binding protein (drSSB) for binding to the freely exposed ssDNA, and efficiently displaces drSSB from ssDNA without consuming ATP. drRecO replaces drSSB and dissociates it completely from ssDNA even though drSSB binds to ssDNA approximately 300 times more strongly than drRecO does. We suggest that drRecO facilitates the loading of RecA onto drSSB-coated ssDNA by utilizing a small drSSB-free space on ssDNA that is generated by the fast diffusion of drSSB on ssDNA.

**\*For correspondence:**
seongkim@snu.ac.kr (SKK);
namkilee@snu.ac.kr (NKL)

**Competing interests:** The authors declare that no competing interests exist.

## Introduction

DNA double-strand breaks (DSBs) are considered the most deleterious of DNA lesions, underlying various classes of damage that threaten genomic integrity among all living organisms. RecA-mediated homologous recombination (HR) is one of the major pathways for repairing DSBs, and a biological mechanism that is nearly universal from bacteria to humans (*Kaniecki et al., 2018*; *Kowalczykowski, 2015*; *San Filippo et al., 2008*). It involves three steps: end processing of the damaged DNA, presynaptic formation of RecA on single-stranded DNA (ssDNA), and RecA-mediated strand invasion of the intact homologous chromosome as a template for further repair processes. In the first step, the exonucleases and helicases generate a small region of exposed ssDNA in the 3′ overhang, which is immediately protected by ssDNA-binding protein (SSB). Strong interactions between ssDNA and SSB ($K_d$ ~pM) hinder RecA assembly on ssDNA by blocking the initial binding of RecA to ssDNA (*Inoue et al., 2011*; *Kowalczykowski, 2000*).

Prokaryotes have two major pathways that load RecA onto SSB-coated ssDNA, that is the RecBCD and RecFOR pathways (*Chung and Li, 2013*; *Cox, 2007*; *Kowalczykowski et al., 1994*; *Yu-Chin et al., 1994*). In *Escherichia coli* (*E. coli*), the RecBCD system predominantly repairs damaged dsDNA, whereas the RecFOR pathway is mainly used for ssDNA gap repair and remains a backup machinery for repairing DSBs (*Chung and Li, 2013*; *Kowalczykowski et al., 1994*; *Lloyd and Buckman, 1985*; *Yu-Chin et al., 1994*). An analysis of bacterial genomes showed that only ~50% of bacteria contain *recBCD*, while 95% of bacterial genomes contain *recO* (*Garcia-Gonzalez et al., 2013*). This finding suggests that the RecOR (or RecFOR) pathway is used more frequently than the RecBCD

pathway for RecA-mediated HR in bacterial species (*Kowalczykowski, 2015*; *Rocha et al., 2005*). Furthermore, the RecFOR pathway resembles the HR system found in humans, which is composed of Rad52 and BRCA2, more closely than does the RecBCD pathway (*Kowalczykowski, 2015*). However, the mechanistic details of RecFOR in the RecA nucleation process are largely unknown in all bacteria. RecFOR-mediated RecA nucleation is the rate-determining step of the early HR process because the RecFOR machinery must overcome the strong interactions between SSB and ssDNA in order to enable RecA loading on ssDNA (*Bell et al., 2015*; *Bell et al., 2012*; *Hobbs et al., 2007*). RecO is believed to play a key role in this step because RecO is able to interact with SSB, ssDNA, and dsDNA simultaneously (*Handa et al., 2009*; *Hobbs et al., 2007*; *Inoue et al., 2011*).

Here, we investigated the detailed molecular mechanism of *D. radiodurans* (DR) RecO (drRecO) in SSB displacement from ssDNA at the single-molecule level. DR is the toughest bacterium known, with outstanding resistance to ionizing radiation and DNA damage-inducing reagents (*Blasius et al., 2008*; *Cox and Battista, 2005*; *Slade and Radman, 2011*), which are the main causes of DSBs, a form of fatal biological damage at the genomic level (*Yokoya et al., 2002*). DR can survive doses of gamma radiation as high as approximately 15,000 Gy, whereas the minimal lethal doses of gamma radiation for *E. coli* and humans are 60 Gy and 5 Gy, respectively (*Battista, 1997*; *Minton, 1994*; *Moseley and Mattingly, 1971*). Previous studies found that DR does not have any distinctive mechanism for gene protection that is distinct from those of other bacteria (*Battista et al., 1999*). However, the entire repair process for DSBs is notably more efficient in DR than in other organisms; DR is able to restore hundreds of copies of damaged double strands in an hour (*Daly et al., 1994*; *Lin et al., 1999*). Despite this remarkable efficiency, the quantities and compositions of the proteins involved in the repair machinery of DR are not distinct from those of other bacteria (*Lin et al., 1999*; *White, 1999*). Thus, how DR achieves DSB repair with such high efficiency remains unclear. Interestingly, there is no evidence for the existence of RecB and RecC, the key proteins in the RecBCD process, in DR. RecF, RecO, and RecR, which play minor roles in *E. coli*, have, however, been identified by whole-genome sequencing as components of a major DSB repair system in DR (*Makarova et al., 2001*; *White, 1999*). Crystal structures and molecular dynamics simulations revealed conformational changes in the drRecOR complex upon single- and double-stranded DNA binding, suggesting that drRecO may lead to drSSB displacement from ssDNA (*Radzimanowski et al., 2013*; *Timmins et al., 2007*).

In this work, we directly observed competitive binding events between drRecO and drSSB on the same ssDNA in real time using single-molecule fluorescence resonance energy transfer (smFRET). drRecO eventually overcomes the strong binding between drSSB and ssDNA and completely dissociates drSSB from ssDNA. We found that drRecO effectively removes drSSB from ssDNA without consuming ATP, despite drSSB binding to ssDNA 300 times more strongly than drRecO does. drRecO binds to ssDNA in a two-step binding mode using its two DNA binding sites, and this two-step binding mode plays a key role in removing drSSB from ssDNA without using ATP. The first binding between drRecO and ssDNA occurs in the drSSB-free space of ssDNA and generates a heterotrimer of the drSSB-ssDNA-drRecO complex as an intermediate state. The second binding between drRecO and ssDNA induces the dissociation of drSSB from ssDNA by facilitating a conformational change in ssDNA.

## Results

### Observation of drSSB displacement from ssDNA by drRecO in solution

We introduced the alternating laser excitation fluorescence resonance energy transfer (ALEX-FRET) technique, which observes the FRET values of a freely diffusing ssDNA labeled with a donor and an acceptor dye (*Figure 1A*; *Kapanidis et al., 2004*; *Lee et al., 2005*). A partial duplex DNA containing an ssDNA overhang composed of 70-nucleotide (nt) poly-dT (dT70) was introduced to monitor the conformation of 70-nt ssDNA. A donor (Cy3B) and an acceptor (Cy5) dye were attached near the two ends of the ssDNA overhang (*Figure 1A*). The binding of drSSB and/or drRecO to a freely diffusing ssDNA was monitored by the change in the FRET value of the ssDNA (*Figure 1B–E*). It is well known that drSSB specifically binds to ssDNA using two oligonucleotide/oligosaccharide-binding (OB) fold domains per monomer and functions as a stable homodimer (*Kozlov et al., 2010*; *Witte et al., 2005*; *Zhou et al., 2011*). drSSB is wrapped by ssDNA longer than 50 nt, and the

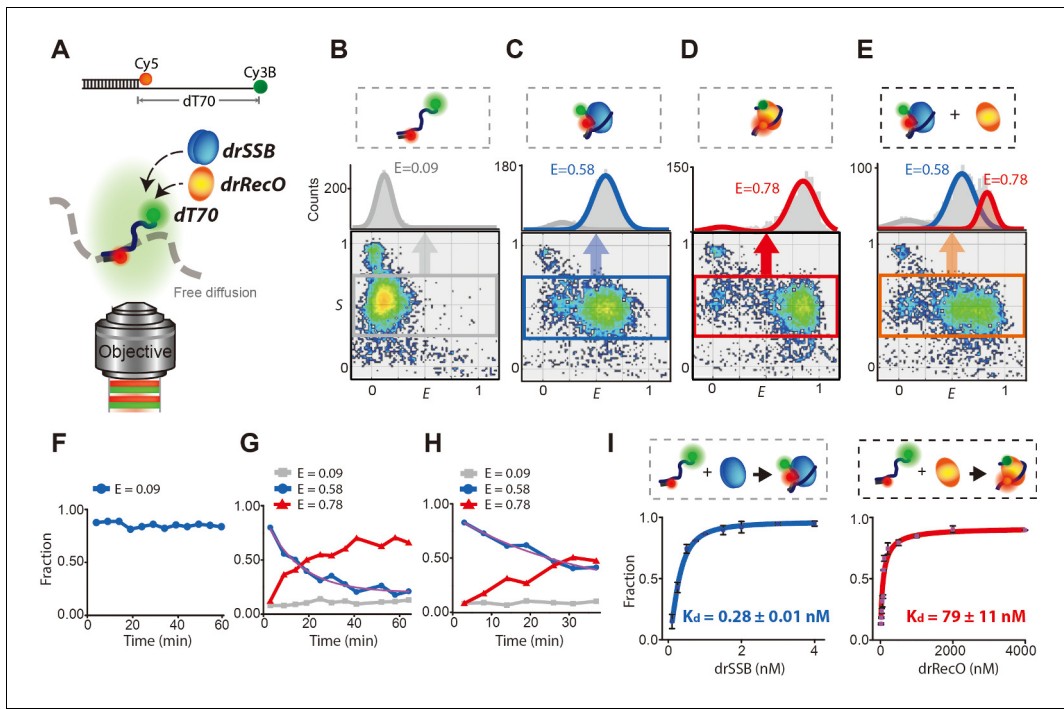

**Figure 1.** Displacement of drSSB from individual diffusing ssDNA by drRecO. (**A**) Schematic illustration of the single-molecule FRET measurement for freely diffusing drSSB- and/or the drRecO–ssDNA complex by ALEX-FRET. Cy3B (donor) and Cy5 (acceptor) were attached to the ends of ssDNAs composed of 70 nucleotides (dT70) and a ssDNA–dsDNA junction, respectively. The donor and acceptor were 70 nt apart, which enabled the monitoring of drSSB or drRecO binding to ssDNA. (**B–E**) Two-dimensional E-S graphs obtained by ALEX-FRET. E-S graphs are used to select ssDNAs labeled with both donor and acceptor dyes (the gray box in panel **B**). The 1D FRET histogram of the selected ssDNAs is plotted above the E-S graph: (**B**) dT70, (**C**) dT70 + drSSB and (**D**) dT70 + drRecO. (**E**) After the formation of the drSSB–dT70 complex was allowed to proceed for 10 min, 500 nM drRecO were added to the mixture. Averaged E values were obtained by fitting 1D FRET histograms with Gaussian function. E = 0.09 (gray lines), 0.58 (blue lines) and 0.78 (red lines) for dT70, drSSB–dT70, and drRecO–dT70, respectively, were obtained. (**F–H**) Time-dependent population changes of dT70 (E = 0.09), drSSB–dT70 (E = 0.58), and drRecO–dT70 (E = 0.78). The final concentrations of drSSB and dT70 were 200 nM and 100 pM, respectively. In the absence of drRecO, drSSB–dT70 remained stable for more than 60 min (**F**). The addition of (**G**) 200 nM and (**H**) 100 nM drRecO into the preassembled dT70–drSSB complex facilitated drastic changes in the population ratios. The exchange rates (purple lines) were $1.10 \times 10^{-3}$ $s^{-1}$ and $5.02 \times 10^{-4} s^{-1}$ for panels (**G**) and (**H**), respectively (see 'Materials and methods'). (**I**) Measurement of the apparent dissociation constants ($K_d$) of drSSB–ssDNA and drRecO–ssDNA. The ALEX-FRET method allowed the measurement of the $K_d$ values of drSSB–ssDNA and drRecO–ssDNA (see 'Materials and methods'). From the half-saturation, the $K_d$ values were found to be $0.28 \pm 0.01$ nM and $79 \pm 11$ nM for drSSB–ssDNA and drRecO–ssDNA, respectively.

The online version of this article includes the following figure supplement(s) for figure 1:

**Figure supplement 1.** Schematic illustration of ALEX microscope setup and data interpretation.
**Figure supplement 2.** Dissociation kinetics of drSSB–ssDNA.

---

drSSB–ssDNA complex is very stable under high salt conditions (>200 mM) (*Kozlov et al., 2010*; *Witte et al., 2005*).

ALEX-FRET detects the FRET efficiency (E) of a freely diffusing molecule in solution, bypassing the surface immobilization of molecules (*Figure 1—figure supplement 1*). The result of ALEX-FRET is presented in a two-dimensional (2D) E-S graph (*Figure 1B–E*). Each dot in the E-S graph represents a detected single molecule. ssDNA labeled with both a donor and an acceptor dye is selected from the E-S graph (the gray box in *Figure 1B*), and then the 1D FRET histogram of the selected ssDNA is plotted on the upper side of the E-S graph. In the absence of drSSB and drRecO, the averaged E value of dT70 was 0.09 (*Figure 1B*). When we added drSSB to the solution containing dT70, the FRET histogram became heterogeneous, showing two populations at E = 0.09 and E = 0.58,

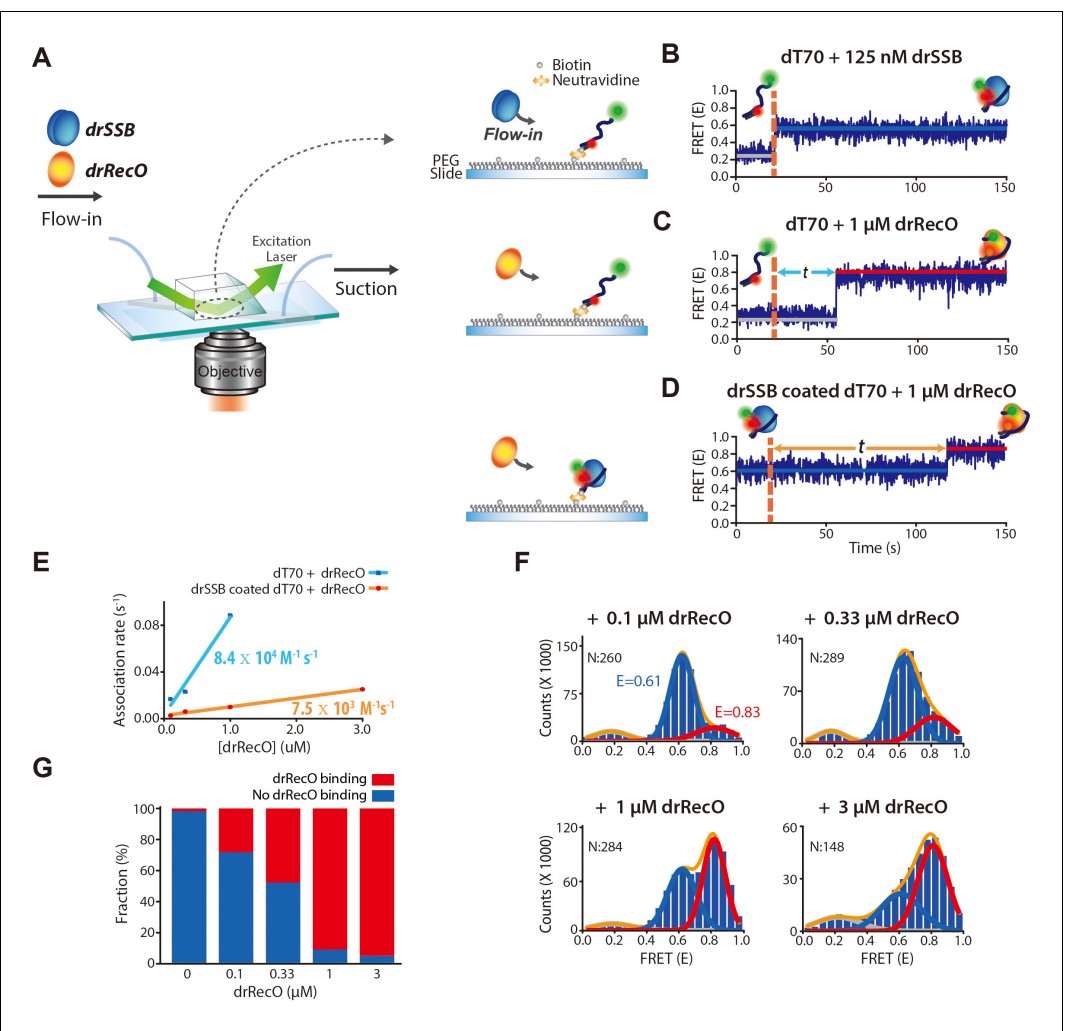

**Figure 2.** Real-time observation of drSSB displacement from immobilized ssDNA by drRecO. (**A**) Schematic descriptions of TIRF microscopy and the experimental scheme for the real-time measurement of drRecO-mediated drSSB displacement from ssDNA. dT70 ssDNA was tethered to the glass surface using the NeutrAvidin–biotin interaction. Then, drSSB and drRecO were allowed to flow into the reaction chamber. All time traces were obtained for 5 min. (**B**) Single-molecule FRET time trace of dT70 binding to drSSB. drSSB (125 nM) was injected into the flow channel (the dashed line denotes the injection time of drSSB), where dT70 was immobilized on the surface. The FRET efficiency increased from ~0.2 (bare ssDNA) to ~0.6 (drSSB–ssDNA complex) in the time trace. The FRET peak centered at 0.62 for the cumulated histogram of all FRET efficiencies (*Figure 2—figure supplement 1A*). The FRET change occurred immediately after the addition of drSSB. (**C**) Single-molecule FRET time-trace of dT70 binding to drRecO. First, 1 μM drRecO was injected into the flow channel (the dashed line denotes the injection time of drRecO), and dT70 was immobilized on the surface. The FRET efficiency was increased from ~0.2 to ~0.8 (drRecO–ssDNA complex). The FRET peak was centered at 0.83 in the FRET distribution for all molecules (*Figure 2—figure supplement 1B*). The time required for drRecO to bind ssDNA (*t*) is longer than that for drSSB. (**D**) Single-molecule FRET time-trace of drSSB displacement from dT70 by drRecO. 125 nM drSSB was injected into the flow channel and incubated for 10 min for the formation of the drSSB–dT70 complex. Then, 1 μM drRecO was injected into the flow channel (the dashed line). During the injection of drRecO, unbound drSSB was washed out. The FRET efficiency increased from ~0.6 to ~0.8, which denote drSSB–dT70 and drRecO–dT70, respectively. Two populations were distinctively separated in the cumulative histogram of all FRET efficiencies (*Figure 2—figure supplement 1C*). (**E**) Comparison of the association kinetics of drRecO to dT70 and to the drSSB–dT70 complex. The time required for drRecO to bind ssDNA is marked as *t* in panels (**C**) and (**D**). From the average *t*, the association rates between drRecO and dT70 and between drRecO and drSSB-coated dT70 were obtained. As the concentration of drRecO increased, the association rates increased. The linear fits yielded the first-order rate constants of $8.4 \times 10^4 \ M^{-1} \cdot s^{-1}$ and $7.5 \times 10^3 \ M^{-1} \cdot s^{-1}$ for dT70 and drSSB-dT70, respectively. (**F**) 1D FRET histograms of drSSB displacement by drRecO at different concentrations of drRecO. The

*Figure 2 continued on next page*

*Figure 2 continued*

cumulative histograms of all FRET efficiencies were obtained by applying different drRecO concentrations to drSSB-coated dT70 in panel (**D**). N denotes the number of time traces accumulated. The Gaussian fit for each population was used to find the binding fraction of drRecO–dT70 (red lines at E = 0.83). The population at E = 0.61 denotes drSSB–dT70. (**G**) Fraction of drRecO binding to drSSB–ssDNA complex at various drRecO concentrations. We counted time traces that showed E = 0.8, which indicated the successful replacement of drSSB by drRecO on ssDNA within 5 min of observation time. The number of time traces analyzed is presented in panel (**F**).

The online version of this article includes the following source data and figure supplement(s) for figure 2:

**Source data 1.** Data summary table for the results shown in *Figure 2E*.
**Source data 2.** Data summary table for the results shown in *Figure 2G*.
**Figure supplement 1.** The fluorescence intensity time traces and cumulated FRET histograms of drSSB and drRecO binding to dT70 and drRecO binding to drSSB-coated dT70.

corresponding to free and drSSB-coated dT70, respectively (*Figure 1C*). Then, we tested whether drRecO competitively interacts with drSSB-coated ssDNA. Interestingly, a new population, emerging at E = 0.78, was found upon the addition of drRecO to drSSB–dT70 (*Figure 1E*). Although the structure of the drRecO–ssDNA complex and the binding properties of drRecO to ssDNA have not yet been identified, we confirmed that the population at E = 0.78 corresponded to drRecO–dT70 by a control experiment using drRecO only (*Figure 1D*). To understand how the drRecO–ssDNA complex forms kinetically when drRecO is applied to the drSSB–ssDNA complex, we measured each fraction of ssDNA only, drSSB–ssDNA, and drRecO–ssDNA over time. First, drSSB was preincubated with dT70 for 10 min to form the fully bound drSSB–dT70 state, and then drRecO was added to the mixture (the final concentrations of drSSB, dT70, and drRecO were 200 nM, 50 pM, and 200 nM, respectively). The fraction of drSSB–dT70 at E = 0.58 in the absence of drRecO remained almost constant for more than 60 min (*Figure 1F*). On the other hand, the fraction of drSSB–dT70 (E = 0.58) gradually decreased and the fraction of drRecO–dT70 (E = 0.78) increased in a complementary manner (*Figure 1G*). When the concentration of drRecO was reduced to half of the drSSB concentration (100 nM drRecO and 200 nM drSSB), drSSB was still replaced by drRecO (*Figure 1H*). Furthermore, the exchange rates between drSSB–dT70 and drRecO–dT70 were highly dependent on the concentration of drRecO (*Figure 1G–H*).

It is known that SSB has an exceptionally slow off-rate on ssDNA. In particular, it has been reported that SSB from *E. coli* (ecoSSB), which has a largely conserved structure relative to that of drSSB, stays on ssDNA for more than several hours (*Witte et al., 2005*; *Zhou et al., 2011*). However, the dissociation rate of drSSB on ssDNA has not been reported. Thus, we measured the dissociation kinetics of drSSB–ssDNA at a single-molecule level (*Figure 1—figure supplement 2*). Only 5% of drSSB was dissociated from dT70 for 2 hr. When equal amounts of drSSB and drRecO were applied, half of the drSSB was displaced from ssDNA by drRecO in approximately 15 min (*Figure 1G*). The fast replacement by drRecO compared to the SSB dissociation rate indicates that drRecO actively induces the dissociation of drSSB from ssDNA. In addition, the dissociation constant ($K_d$) of the drRecO–dT70 complex is approximately 300 times larger than that of the drSSB–dT70 complex (*Figure 1I*). Our results suggest that drRecO can bind to drSSB-coated ssDNA despite drSSB interacting with ssDNA approximately 300 times more strongly than drRecO does.

## Real-time observation of drSSB displacement by drRecO using single-molecule FRET

To observe the displacement of drSSB by drRecO in real time, we tethered ssDNA on a glass surface using a NeutrAvidin–biotin interaction, and employed a total internal reflection fluorescence (TIRF) microscope (*Figure 2A*). The apparent E value of the free ssDNA dT70 was approximately 0.2. When drSSB bound to the surface-immobilized dT70, the E value of dT70 increased from ~0.2 to ~0.6, and the drSSB–dT70 complex was stable throughout our observation time (up to 300 s) (*Figure 2B* and *Figure 2—figure supplement 1A and B*). In the same manner, the binding of drRecO to dT70 increased the E value from ~0.2 to ~0.8 (*Figure 2C* and *Figure 2—figure supplement 1C and D*). Then, we added drRecO to the preformed drSSB–dT70 complex. During the injection of drRecO, unbound drSSB was washed out in the surface-tethered experiments: the E value

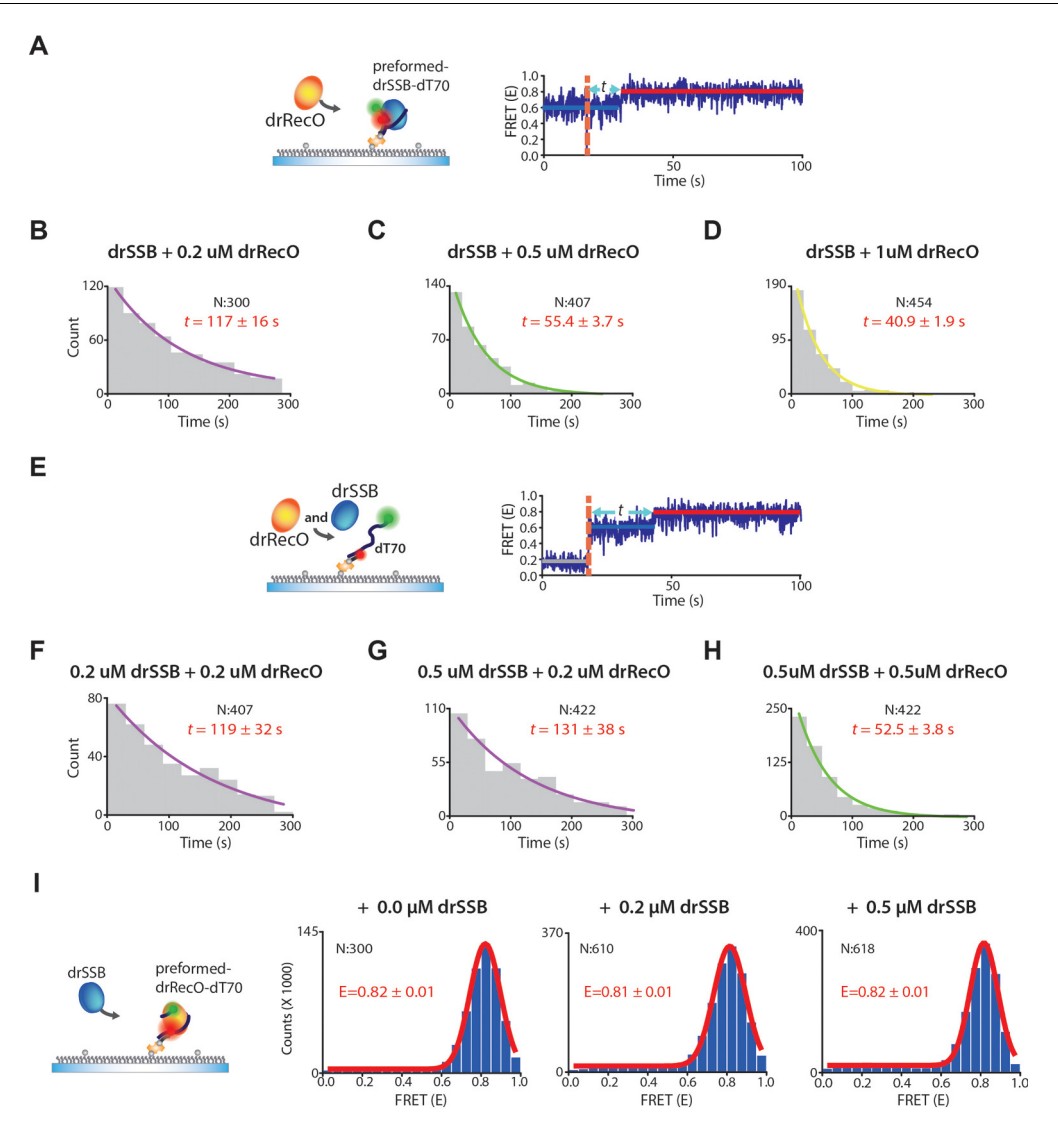

**Figure 3.** The exchange rates by drRecO in the preformed and competitive conditions. (**A**) Schematic illustration of the preformed condition. (**B–D**) The concentration-dependent exchange rates by drRecO in the preformed condition. drSSB was washed out when drRecO was injected. We added 0.2 μM, 0.5 μM, and 1.0 μM drRecO, respectively. (**E**) Schematic illustration of the competitive condition. We added drRecO and drSSB simultaneously. (**F–H**) The concentration-dependent rates of drRecO exchange in the competitive condition. We added (**F**) 0.2 μM drSSB and 0.2 μM drRecO, (**G**) 0.5 μM drSSB and 0.2 μM drRecO SSB, and (**H**) 0.5 μM drSSB and 0.5 μM drRecO, respectively. The exchange rate is primarily affected by the concentration of drRecO, whereas the concentration of drSSB does not notably change the rate. (**I**) Schematic illustration and the FRET histogram at various concentrations of drSSB to confirm whether drSSB displaces drRecO from the preformed drRecO–dT70 complex in real-time TIRF measurement. FRET values of drRecO–dT70 were not affected by drSSB. Thus, drSSB does not displace drRecO from the preformed drRecO–dT70.

The online version of this article includes the following figure supplement(s) for figure 3:

**Figure supplement 1.** Observation of drSSB and drRecO binding to short ssDNAs (dT20, dT30 and dT40) by ALEX-FRET.

increased directly from ~0.6 to ~0.8 (*Figure 2D* and *Figure 2—figure supplement 1E and F*). The direct change in the E value from ~0.6 to ~0.8 implies that drSSB is displaced from dT70 by drRecO and that drRecO forms a complex with dT70. The association rate of drRecO to dT70 is more than 10 times larger than that of drSSB-coated dT70 (*Figure 2E*), which is consistent with previous work

showing that SSB limits the interaction between RecO and ssDNA in *E. coli* (*Handa et al., 2009*). As the concentration of drRecO was increased, the fraction of drSSB–dT70 (E ~ 0.6) gradually decreased, while that of drRecO–dT70 (E ~ 0.8) increased during 4.5 min of reaction time (*Figure 2F and G*).

Then, we measured the exchange (drSSB-displacement) rate by drRecO (*Figure 3A–D*). Again, the exchange rate was enhanced as the concentration of drRecO increased (*Figure 3B–D*). In the surface-tethered experiment, unbound drSSB was washed out during drRecO injection. Then, we tested whether the unbound drSSB could compete with drRecO for binding to drSSB-coated ssDNA (*Figure 3E–H*). We simultaneously added drRecO and drSSB and allowed them to bind competitively to ssDNA. The dotted orange line in the time-trace of *Figure 3E* represents the time point at which 0.2 μM drRecO and 0.2 μM drSSB were injected together. After adding drSSB and drRecO to the surface-tethered dT70, the E value increased immediately from ~0.2 to ~0.6, which was followed by a second increase in the E value from ~0.6 to ~0.8. This indicates that drSSB immediately binds to dT70 and forms the same initial state of the preincubated drSSB–ssDNA complex. This result is consistent with the measurement of the association rate of drSSB as being approximately 10,000 times larger than that of drRecO (*Witte et al., 2005*). Notably, when we added 0.2 μM drRecO in the presence of 0 μM, 0.2 μM, and 0.5 μM drSSB in solution (*Figure 3B,F and G*, respectively), the exchange rates were nearly invariant. However, the exchange rates increased as the concentration of drRecO increased from 0.2 μM to 0.5 μM in the presence of 0.5 μM drSSB (*Figure 3G and H*). The exchange rates were independent of drSSB concentration in solution. These results indicate that unbound drSSB in solution is not a competitor of drRecO for drSSB-coated ssDNA binding and that drRecO is able to displace drSSB from ssDNA regardless of whether unbound drSSB is present in solution.

Next, we investigated whether drSSB removes resident drRecO from dT70 (*Figure 3I*). We incubated 500 nM drRecO with surface-tethered dT70s for 10 min, allowing drRecO to bind fully to dT70. The FRET value once again increased to ~0.8 upon the binding of drRecO to dT70. We then injected drSSB into the drRecO–dT70 complex (free drRecO was washed out during drSSB injection). We observed that the FRET value was maintained in the presence of 0.5 μM drSSB (*Figure 3I*). This result indicates that drSSB cannot remove drRecO from the drRecO–dT70 complex, whereas drRecO removes drSSB from ssDNA. It is possible that drSSB may not displace the resident drRecO from the preformed drRecO–ssDNA complex because drSSB requires a high number of free nucleotides for ssDNA binding compared to drRecO (*Figure 3—figure supplement 1*). drRecO completely dissociates drSSB from ssDNA.

Then, we investigated the details of the function of drRecO in drSSB replacement from ssDNA. On the basis of the suggested mechanisms of SSB removal by RecO in other bacteria, three models are possible (*Figure 4A*). The most intuitive model is that RecO replaces and fully dissociates SSB from ssDNA (Model 1). However, in *Thermus thermophilus* (tt), which is phylogenetically related to DR, ttRecO displaces ttSSB by strongly binding to the DNA, but the unbound ttSSB remains tethered (Model 2) (*Inoue et al., 2008*; *Inoue et al., 2011*). Model 2 could include an intermediate state prior to the full removal of SSB from ssDNA. In *E. coli*, ecoRecO forms a complex with ecoRecR and ssDNA-bound ecoSSB, which is expected to alter the conformation of ecoSSB-bound ssDNA without dissociation (Model 3) (*Umezu and Kolodner, 1994*). Similar to the process in *E. coli*, RecO found in *Mycobacterium smegmatis* (msRecO) does not fully dissociate msSSB from ssDNA. However, msRecO does not bind directly to the C-terminus of msSSB, where ecoRecO binding is well known (*Ryzhikov et al., 2014*). As with Model 2, it remains possible that Model 3 is an intermediate state that occurs prior to Model 2 and eventually Model 1, in which SSB is fully dissociated from ssDNA.

In other bacteria, it seems to be common that SSB remains in the RecO–ssDNA complex as an intermediate or final state. Thus, we first investigated whether drSSB completely dissociates from ssDNA or whether it remains tethered on ssDNA when replaced by drRecO. We measured the colocalization between fluorescently labeled drSSB (Cy5–drSSB) and ssDNA (Cy3–dT70) after adding drRecO using a dual-color TIRF microscope (*Figure 4B* and *Figure 4—figure supplement 1*). A single point mutation was introduced into drSSB to allow fluorescent labeling (G120C). We measured the fraction of colocalization between Cy5–drSSB and Cy3–dT70 under various concentrations of drRecO (*Figure 4C and D*). Without drRecO, 72% of Cy5–drSSB and Cy3–dT70 were colocalized for 4.5 min. Because drSSBs were not dissociated from ssDNA during the observation time (*Figure 1—figure supplement 2*), most of the 28% non-colocalized population came from the

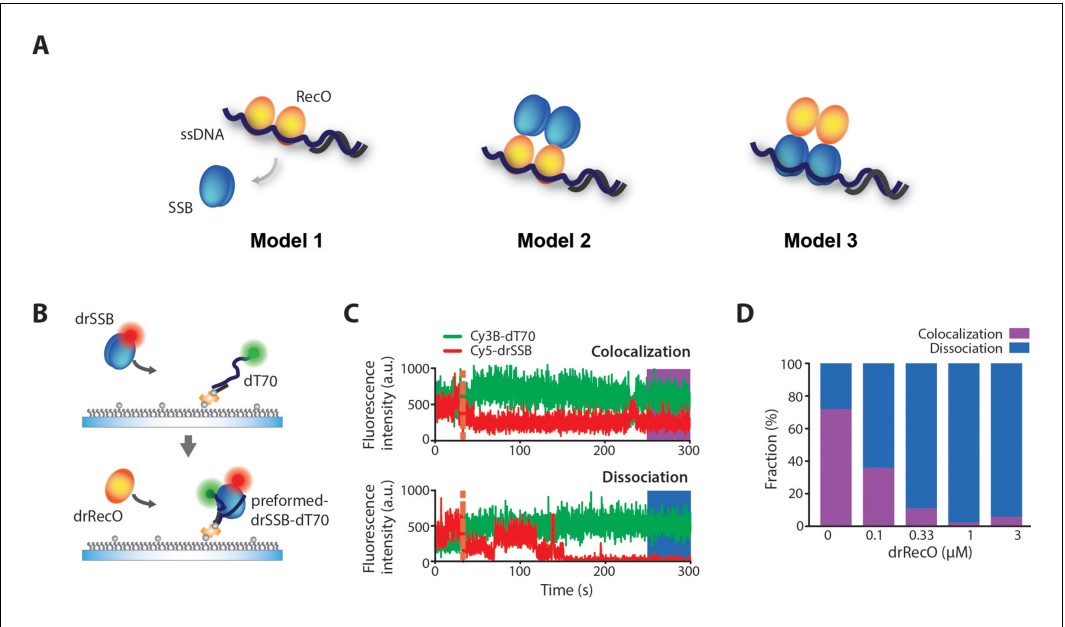

**Figure 4.** Colocalization of drSSB and ssDNA after drSSB displacement by drRecO. (**A**) Three models of SSB displacement from ssDNA by RecO. In Model 1, SSB is completely dissociated from ssDNA by RecO. In Model 2, RecO displaces SSB, but SSB remains on the complex by binding to RecO. In Model 3, RecO does not dissociate SSB from ssDNA but alters the motion of SSB. (**B**) A schematic representation of the drRecO-dependent colocalization measurement of Cy5–drSSB and Cy3B–dT70 using two-color TIRF microscopy. Cy5–drSSB was preincubated with surface-tethered Cy3b–dT70 to form drSSB–dT70 complex. Then, drRecO was flowed through the reaction chamber. During the injection of drRecO, unbound drSSB was washed out. (**C**) Representative time trajectories of the colocalization of drSSB and dT70 (top panel) and the dissociation of drSSB from dT70 by drRecO (bottom panel). The orange dashed line indicates the time of drRecO injection. The ratio of colocalization was obtained from the trajectories at 4.5 min after the addition of drRecO. (**D**) Colocalization fractions of drSSB and dT70 at various concentrations of drRecO. Higher concentrations of drRecO induced more dissociation between drSSB and dT70, implying that drRecO induces the dissociation of drSSB from ssDNA, as shown in Model 1.

The online version of this article includes the following source data and figure supplement(s) for figure 4:

**Source data 1.** Data summary table for the results shown in *Figure 4D*.
**Figure supplement 1.** Colocalization measurement of fluorescently labeled drSSB and ssDNA.
**Figure supplement 2.** Measurement of photo-bleaching and blinking on TIRF imaging.

---

photobleaching of acceptor dyes (*Figure 4—figure supplement 2*). When we applied various concentrations of drRecO, the colocalized fraction of drSSB–ssDNA decreased significantly after 4.5 min of reaction time (*Figure 4D*). When more than 1 μM drRecO was applied, a minor fraction of colocalization between drSSB–ssDNA was observed. These results imply that drSSB was completely dissociated from ssDNA by drRecO (Model 1). This is a unique feature of DR compared with other bacteria, in which SSB remains in the RecO–ssDNA complex.

## Molecular mechanism of drSSB displacement from ssDNA by drRecO without consuming ATP: observation of an intermediate state in the process of drSSB displacement

Other proteins that displace SSB from ssDNA, such as translocase and helicase, use ATP to overcome the strong binding affinity between SSB and ssDNA (*Lee et al., 2013*; *Lohman et al., 2008*; *Park et al., 2010*). However, we found that drRecO can remove drSSB from ssDNA without consuming ATP, even though the binding (dissociation constant) of drRecO to ssDNA is more than two orders of magnitude weaker than that of drSSB. Thus, we further investigated the detailed mechanism by which drRecO displaces drSSB without using ATP.

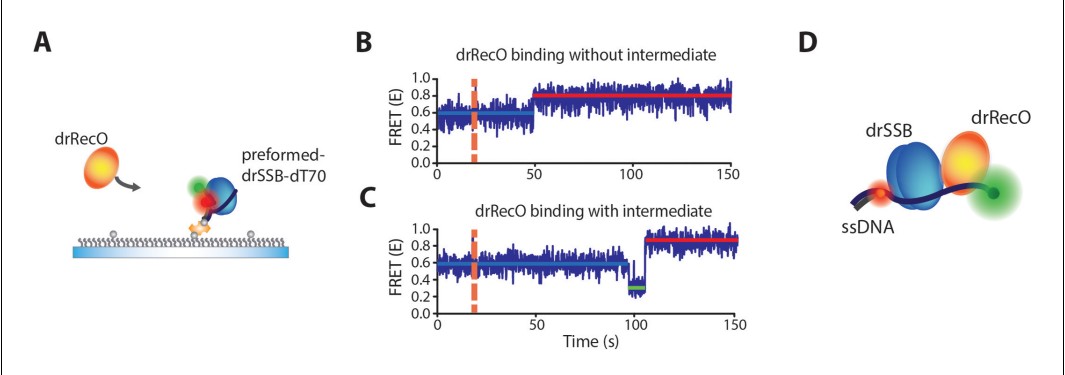

**Figure 5.** Observation of the intermediate state during drSSB displacement from ssDNA by drRecO. (A) Schematic illustration of the binding of drRecO to the preassembled drSSB–dT70 complex. The same measurement is presented in *Figure 2D*. (B, C) FRET time traces of the binding of drRecO to the preassembled drSSB–dT70 complex. The orange dashed lines indicate the time at which 1 µM drRecO was applied. Most time traces exhibited no intermediate state, as shown in panel (B). Approximately 2% of the time traces presented an intermediate state with E = ~0.2, as shown in panel (C). (D) A possible model for the intermediate state of a trimer (drRecO–ssDNA–drSSB) formed before drSSB displacement from the complex. The low E value (~0.2) indicates that ssDNA has a relatively stretched form.

When we applied drRecO to the drSSB–dT70 complex (*Figure 2D*), most of the molecules showed an increase in the E value from ~0.6 to ~0.8 (*Figure 5B*). Interestingly, some dT70 molecules occasionally presented an intermediate state with E ~ 0.2 before the transition to E ~ 0.8 (*Figure 5C*). The population of the time traces showing the intermediate state was smaller than 2%. Because drSSB typically maintains stable binding to dT70, the intermediate state could be the trimer of drRecO–dT70–drSSB that is formed before the dissociation of drSSB from dT70 (*Figure 5D*).

## Real-time observation of drRecO binding to various lengths of ssDNA

Then, we further investigated the nature of the intermediate state, potentially a drRecO–ssDNA–drSSB trimer, to understand the detailed mechanism of drSSB displacement at the molecular level. If the intermediate state is the heterotrimer, as we suggested, the stability of the intermediate state would be affected by the length of ssDNA. For example, if the ssDNA is too short, there would be no room for the simultaneous stable binding of drRecO and drSSB, which would affect the dynamics of drSSB replacement by drRecO.

Prior to investigating drSSB displacement by drRecO on various lengths of ssDNA, we examined the binding modes of drRecO on ssDNA (*Figure 6*). When we applied 20-nt and 30-nt ssDNAs, a FRET change was observed in response to drRecO binding (*Figure 3—figure supplement 1*). However, the binding was not stable. When the ssDNA length was longer than 40 nt, however, most of the ssDNAs were in a high FRET state, reflecting the formation of stable complexes with drRecO (*Figure 3—figure supplement 1*). These results suggest that at least 40 nt is required for stable binding between ssDNA and drRecO.

Thus, we used dT40, dT50, and dT60 along with dT70 to observe drRecO binding on ssDNA in real time (*Figure 6*). When dT70 was used, a single-step FRET change from E ~ 0.6 to E ~ 0.8 was observed for most of the time traces (*Figure 2C*). In line with the results of dT70, most of the dT40, dT50, and dT60 molecules also presented a single-step FRET transition to E ~ 0.8 by the binding of drRecO (*Figure 6A*, upper panels). Interestingly, a substantial fluctuation in the E value was observed for dT40, dT50, and dT60 before the transition to E ~ 0.8 (the purple boxes in the time traces of *Figure 6A*). This state of FRET fluctuation is distinctive compared with the FRET fluctuation of bare ssDNA (*Figure 6—figure supplement 1*). This FRET fluctuation occurred before the formation of the fully bound drRecO–ssDNA complex, represented by the high E values. After drRecO reached the fully bound state, the FRET efficiency was stably maintained over 5 min (*Figure 6A*, upper panels). The time (*t* in *Figure 6A*, the upper panels) required to reach the fully bound state became longer as the length of the ssDNA decreased (*Figure 6B* and *Figure 6—figure supplement 2*).

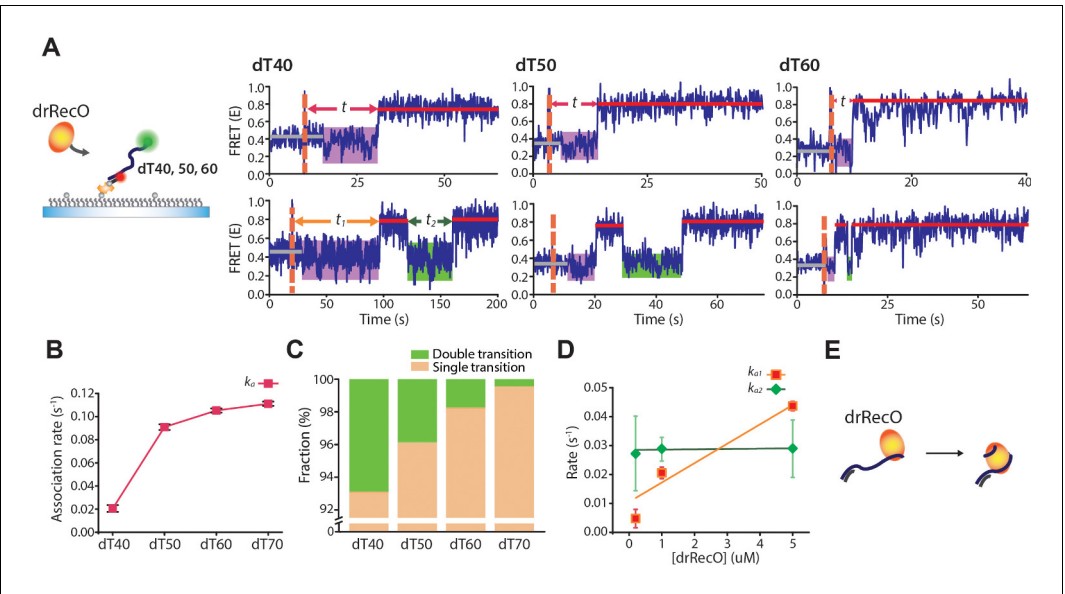

**Figure 6.** Real-time observation of drRecO binding to various lengths of ssDNA. (A) Schematic illustration and representative FRET time trajectories for drRecO binding to dT40 (left), dT50 (middle), and dT60 (right). The orange dashed lines denote the time of drRecO injection. Average FRET values for ssDNA-only species and the fully bound drRecO–ssDNA complexes are indicated with gray and red solid lines. The substantial FRET fluctuations are indicated by the purple boxes. The time required to reach the fully bound state is indicated by $t$ in the upper panels. The lower panels present FRET time traces with the second low FRET state after forming the fully bound drRecO–ssDNA complex (the green boxes). The dwell time of the second low FRET state is marked as $t_2$. (B) The association rates between drRecO and ssDNA ($k_a$) depending on the length of ssDNA. $k_a$ was obtained from the distribution of $t$ in panel (A) (*Figure 6—figure supplement 2B*). (C) The fraction of time traces having multiple FRET transitions depending on the length of ssDNA. (D) The rates of the first and second transition to high FRET depending on drRecO concentration for drRecO binding to dT40. The distributions of the dwell times $t_1$ and $t_2$ were obtained from the single-molecule dwell time analysis. The transition rate was calculated as the reciprocal of the average life-time from a single-exponential fit for the dwell-time distribution. (E) Model of drRecO binding to ssDNA. drRecO binds to ssDNA by the sequential two-step binding mode.

The online version of this article includes the following source data and figure supplement(s) for figure 6:

**Source data 1.** Data summary table for the results shown in *Figure 6B*.
**Source data 2.** Data summary table for the results shown in *Figure 6C*.
**Source data 3.** Data summary table for the results shown in *Figure 6D*.
**Figure supplement 1.** FRET distributions of 40-nt ssDNA and the intermediate state of dT40-drRecO complex.
**Figure supplement 2.** drRecO binding to various lengths of ssDNA by TIRF measurement.

*Leiros et al. (2005)* reported that drRecO interacts with both ssDNA and dsDNA by forming an ion pair between its positively charged residues and the negatively charged backbone of DNA, and they identified two DNA-binding sites in drRecO. The DNA-binding sites are the N-terminal OB fold domain and the positive ridge of the C-terminal. They suggested that these regions were positioned on either end of drRecO and were equally well suited to interact with DNA (*Figure 6—figure supplement 2C*; *Leiros et al., 2005*). The two DNA-binding sites of drRecO may explain the FRET fluctuation (*Figure 6A*, upper panels) and the association rates (*Figure 6B*). As there are two DNA-binding sites in drRecO positioned on both ends of drRecO, it is possible for one of the two ends to bind with ssDNA first, before the other end binds to the ssDNA sequentially. The state with substantial fluctuation of the E values (the purple boxes in *Figure 6A*) occurs when one end of drRecO binds to ssDNA before forming the fully bound complex. In this two-step binding scheme, a shorter ssDNA should have difficulty completing the second binding. As a result, the binding rate of ssDNA becomes slower as the length of ssDNA decreases. This explanation is further supported by RecO mutants that have a mutation on one of the DNA-binding sites, which will be shown later.

Another interesting finding is that a minor fraction of ssDNAs showed multiple FRET transitions, that is after reaching the fully bound state (E ~ 0.8), the E value dropped to the fluctuation state and

then increased to ~0.8 again (*Figure 6A*, bottom panels). These multiple FRET transitions were observed for ssDNAs of all lengths, except for dT70, and the population increased as the length of the ssDNA became shorter (*Figure 6C*). As dT40 showed multiple FRET transitions, we analyzed the rates of the first and second transitions to high FRET. We found that the first transition rate (obtained from the dwell time, $t_1$) was dependent on the drRecO concentration (*Figure 6D*, the red squares), whereas the second transition rate (obtained from $t_2$) was independent of the drRecO concentration (*Figure 6D*, the green diamonds). Thus, the second low-FRET state (marked by the green boxes in *Figure 6A*, the bottom panels) appears because drRecO loses ssDNA binding from one of its two DNA-binding sites. As the length of ssDNA decreases, the probability of losing ssDNA binding increases. Taken together, our results indicate that drRecO binds to ssDNA by a sequential two-step binding mode (*Figure 6E*).

## drRecO mutants reveal the binding mode of drRecO to ssDNA using its two DNA-binding sites

We showed that drRecO sequentially binds to ssDNA using two binding sites (*Figure 6*): the first binding of drRecO induces a partial stretching of ssDNA and the second binding is required to wrap ssDNA to achieve a high E value. To confirm this proposition, we prepared two drRecO mutants in the N-terminal and C-terminal regions, K35E/R39E drRecO and R195E/R196E drRecO, which were designed to have a lower binding affinity to ssDNA. Previous studies reported that drRecO binds to ssDNA and dsDNA with ionic interactions, in which its polar and positively charged residues contact the negatively charged phosphate backbone of DNA (*Leiros et al., 2005*). The K35 and R39 sites are positioned on the OB fold region of its N-terminal domain; thus, they are essential for binding to DNA. In the same manner, the R195 and R196 sites are essential for ssDNA binding to the C-terminal domain of drRecO.

We first tested the binding ability of drRecO mutants to ssDNA dT70 using ALEX-FRET (*Figure 7A*). Compared with the wild-type (wt)-drRecO, the population of the high-FRET state (E = 0.77), the fully wrapped drRecO–dT70, was reduced considerably for both drRecO mutants (*Figure 7A*, the red lines). The reduction was more significant for R195E/R196E drRecO. These results show that both mutants have weaker ability to bind to ssDNA than does wt-drRecO. Interestingly, the drRecO mutants showed a low-FRET state (E = 0.04), whose E value was slightly smaller than that of bare ssDNA dT70 (E = 0.09) (*Figure 7A*, the dotted gray line for bare dT70 and the dotted green line for the low-FRET state of drRecO mutants). The appearance of this low-FRET state was more clearly observed when we used dT40, the shorter length of ssDNA (*Figure 7B*). The E value of bare dT40 was 0.32. When wt-drRecO bound to dT40, the E value increased to 0.72. However, when the drRecO mutants were studied, a new peak at E = 0.03 appeared as a major population, and no high-FRET state (E = 0.72) was observed. The observation that the E values of the drRecO mutant–ssDNA complex are smaller than those of bare ssDNAs indicates that ssDNA interacting with drRecO mutants has a stretched conformation compared with that of bare ssDNA. Because of the short length of dT40, the drRecO mutants failed to bind both sides of dT40. As a result, no high-FRET state was observed (*Figure 7B*, the orange dotted line). When dT40 was used, even wt-drRecO showed a minor population at E = 0.03 (*Figure 7B*). This state occurs when wt-drRecO binds to dT40 using only one DNA-binding site. This result implies that if the ssDNA is too short, drRecO binds the ssDNA using only one DNA-binding site. The binding of wt-drRecO to ssDNA via one binding site induces the stretched form of ssDNA.

All our results confirm that the first binding of drRecO induces partial stretching of ssDNA with a low E value and that the second binding is required to wrap ssDNA to obtain a high E value.

## drSSB displacement from ssDNA by drRecO occurs more frequently via the low-FRET intermediate state for short ssDNAs

Since the binding of drRecO to ssDNA is heavily dependent on the length of the ssDNA (*Figure 6*), we investigated whether the length of ssDNA is also an important factor in the displacement of drSSB from ssDNA by drRecO. drSSB induced a high-FRET state (E ~ 0.7–0.8) with dT40, dT50, dT60 and dT70, and the association between drSSB and ssDNA occurred too fast to be resolved in our TIRF measurement (*Figure 8A* and *Figure 8—figure supplement 1*). When drSSB formed a complex with ssDNA, the complex was very stable during our observation time for all lengths of ssDNA. We

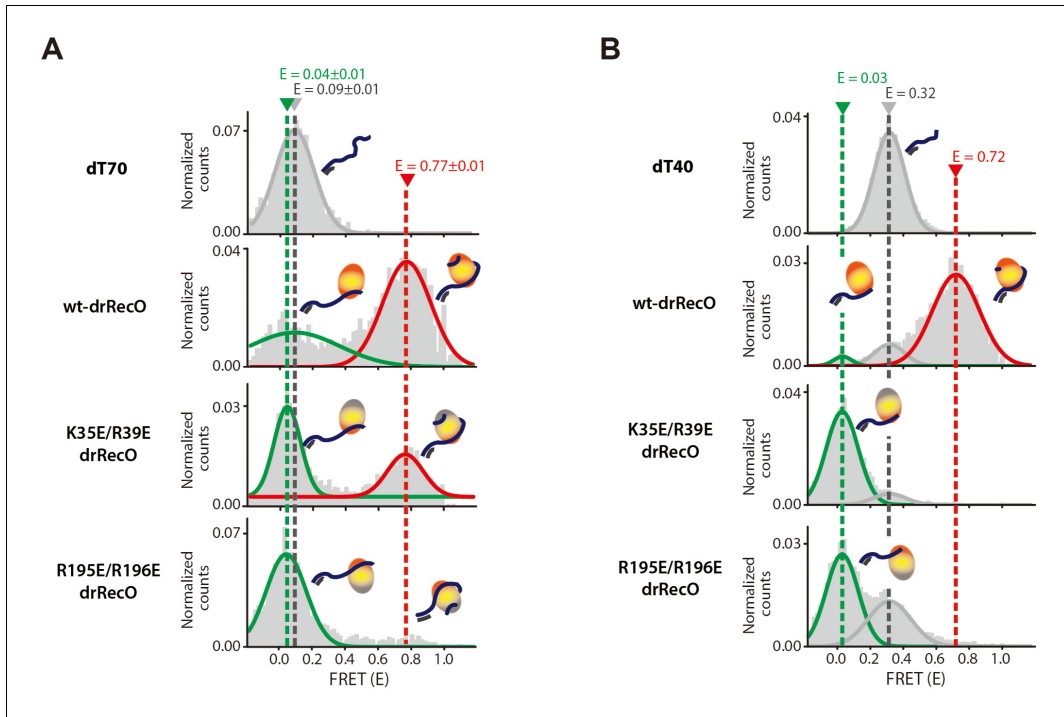

**Figure 7.** The binding modes of drRecO mutants to ssDNA. (**A**) 1D FRET histograms for dT70 only, wt-drRecO–dT70, K35E/R39E drRecO (N-terminal mutation)–dT70, and R195E/R196E drRecO (C-terminal mutation)–dT70, obtained by ALEX-FRET. One of the ssDNA-binding sites was mutated in K35E/R39E drRecO and R195E/R196E drRecO. The gray, green, and red lines represent bare dT70 (E = 0.09), partially bound drRecO–dT70 (E = 0.04), and fully bound drRecO–dT70 (E = 0.77), respectively. (**B**) 1D FRET histograms for dT40 only, wt-drRecO–dT40, K35E/R39E drRecO–dT40, and R195E/R196E drRecO–dT40, obtained by ALEX-FRET. The gray, green, and red lines represent the bare dT40 (E = 0.32), the partially bound drRecO–dT40 (E = 0.03), and the fully bound drRecO–dT40 (E = 0.72), respectively. dT40 showed the partially bound state more clearly because its E value is significantly different from that of bare dT40.

note that the E values of drSSB–ssDNA are similar to those of drRecO–ssDNA for dT50 and dT60 (*Figure 8—figure supplement 1A*). Indeed, when we injected drRecO into the drSSB–ssDNA complex, the E values of the initial and final states for dT50 and dT60 were too close to be distinguished (*Figure 8—figure supplement 1B*). In the case of dT40, however, the E value for drSSB–ssDNA was ~0.8, whereas that of drRecO–ssDNA was ~0.74. Thus, we applied drRecO to the drSSB–dT40 complex (*Figure 8B*). We observed the transition to a low-FRET state more frequently and assigned the low-FRET state as an intermediate state of the drRecO–ssDNA–drSSB trimer (*Figure 8B*). Nearly 60% of dT40 time traces presented more than one FRET transition, which is a much larger percentage than the ~2% of the total dT70 population, as shown in *Figure 5B*. As multiple FRET transitions and a low-FRET state were not observed for drSSB binding to dT40, the low-FRET state corresponds to the intermediate state in the process of displacing drSSB from dT40. Interestingly, 24% of dT40 showed triple or quadruple FRET transitions to the low-FRET state by drRecO (*Figure 8B*). As the unbound drSSB was washed out while injecting drRecO, drSSB should be tethered on ssDNA in the intermediate state to obtain a triple or quadruple FRET transition to the high-FRET state of E ~ 0.8, that is the drSSB–dT40 complex. drRecO should also be tethered on dT40 to maintain the low-FRET state. Otherwise, as shown in *Figure 8A*, dT40 would move immediately to a high-FRET state when it interacts with drSSB. The presence of drRecO may hinder drSSB from fully wrapping dT40 (*Figure 5D*). Thus, the low-FRET intermediate state would be a trimer of drSSB–dT40–drRecO. At the intermediate state, if drRecO dissociates from the trimer, the E value reaches E ~ 0.8. On the other hand, if drRecO successfully removes drSSB, the E value reaches ~0.72 in the time traces presented in *Figure 8B* (the red lines). As the length of ssDNA decreases, drRecO has less chance to

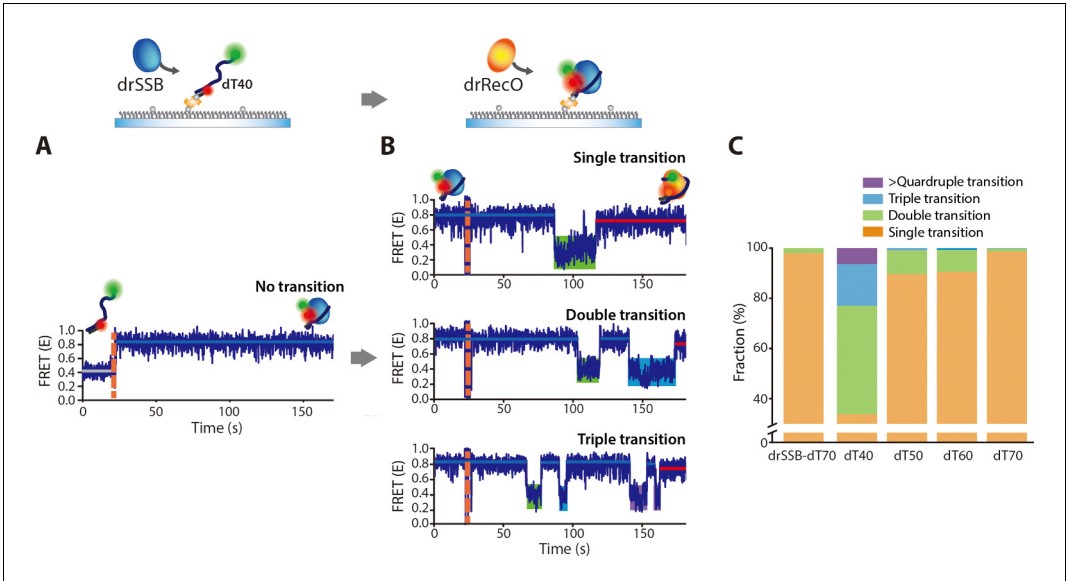

**Figure 8.** drRecO-mediated drSSB displacement depending on the length of ssDNAs. (A, B) Representative time trajectories for (A) drSSB binding to dT40 and (B) drRecO-mediated drSSB displacement from dT40. The drSSB–dT40 complex was formed by adding 125 nM drSSB to surface-immobilized dT40. Immediate transition to high FRET was observed in panel (A). The FRET time trajectories of dT50 and dT60 are presented in *Figure 8—figure supplement 1B*. After the formation of the drSSB–dT40 complex, 1 µM drRecO was added as indicated by the orange dashed lines in panel (B). During the injection of drRecO, unbound drSSB was washed out. The transitions from the drSSB–dT40 complex (the blue lines) to the drRecO–dT40 (red lines) complex occur through the intermediate state of low E value (the green, cyan, and purple boxes). (C) The fraction of the time trajectories having transitions to the low-FRET state. As the length of ssDNA is shorter, multiple transitions to the low-FRET state appear more frequently.

The online version of this article includes the following source data and figure supplement(s) for figure 8:

**Source data 1.** Data summary table for the results shown in *Figure 8C*.
**Figure supplement 1.** The observation of drSSB displacement by drRecO on various lengths of ssDNA.

grab ssDNA with its second binding site. Thus, the low-FRET intermediate state occurs more frequently (*Figure 8C*).

## drRecO displaces drSSB from ssDNA by using a sequential two-step binding mode without consuming ATP

Our results strongly suggest that the intermediate state of the drRecO–ssDNA–drSSB trimer is formed when only one of the binding sites of drRecO binds to ssDNA in the presence of drSSB (*Figure 8*). Then, we applied drRecO mutants to the drSSB–dT70 complex (*Figure 9A–C*). The populations of drSSB dissociated from dT70 by K35E/R39E drRecO and R195E/R196E drRecO were 22% and 7% of the total time traces, respectively, during our observation time of 5 min (*Figure 9C*, the sum of the red and green bars). These values are significantly lower than the 54% obtained for wt-drRecO (*Figure 9C*), which was consistent with the dT70-binding ability observed in *Figure 7A*. Considering that the probability of drSSB dissociation in the absence of drRecO is only 5% for 2 hr (*Figure 1—figure supplement 2*), both drRecO mutants still have the capability to displace drSSB from ssDNA during our observation period.

As the drSSB displacement ability diminished, the time traces showing the intermediate state of E ~ 0.2 (*Figure 9A*) significantly increased to 9% of the total time traces for K35E/R39E drRecO (*Figure 9C*, the green bar). Among the successful drSSB displacement events, 40% of the time traces represented the intermediate state (*Figure 9C*, the green bar/[the green bar + the red bar]). When R195E/R196E drRecO was applied to the drSSB–dT70 complex, 51% of the time traces of successful drSSB displacement events represented the intermediate state (*Figure 9C*, the green bar/[the green bar + the red bar]). Again, double or triple FRET transitions to the low-FRET state by the

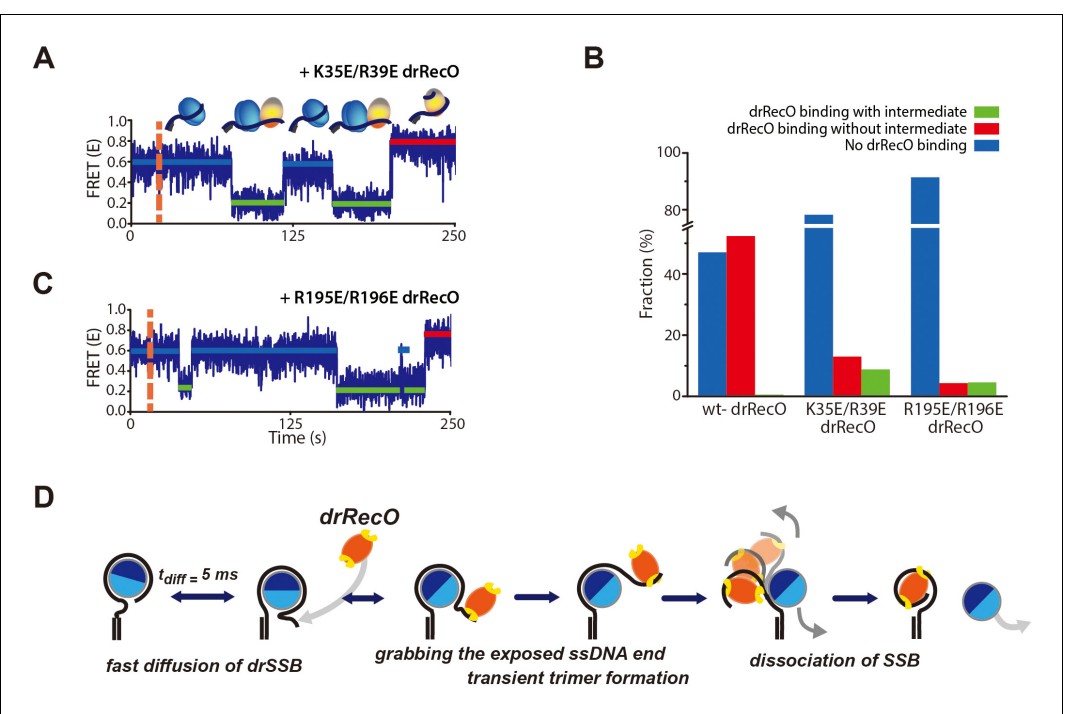

**Figure 9.** Real-time observation of drSSB displacement by drRecO mutants with lower ssDNA-binding affinity.
(A, B) Representative FRET time trajectories for the binding of K35E/R39E drRecO and R195E/R196E drRecO
mutants to the preformed drSSB–dT70 complex. The injection time of the mutants is marked by the orange
dashed lines. During the injection of drRecO, unbound drSSB was washed out. The blue, green, and red solid lines
denote the average FRET values of drSSB–dT70, the heterotrimer of drSSB–dT70–drRecO, and drRecO–dT70,
respectively. Because of the mutation on either end of drRecO, multiple transitions to the intermediate state were
observed more frequently. (C) Fraction of drSSB displacement by drRecO and its mutants. For all measurements,
330 nM drRecO or drRecO mutants were applied to the preformed drSSB–dT70 complex. The blue, red, and
green bars denote the time-trace populations with no drRecO binding (no drSSB displacement), drSSB
displacement without the intermediate state, and drSSB displacement with the intermediate state, respectively.
Mutation at either end of drRecO reduced the proportion of successful drSSB displacement from dT70 but
increased the proportion of time traces showing multiple FRET transitions to the intermediate state. (D) A
proposed model for the displacement of SSB from ssDNA by RecO in DR without consuming ATP.
The online version of this article includes the following source data and figure supplement(s) for figure 9:

**Source data 1.** Data summary table for the results shown in *Figure 9C*.
**Figure supplement 1.** R121A drRecO mutation hinders both drRecO–ssDNA binding and formation of the
intermediate state.

---

drRecO mutants were clearly observed for the drSSB–dT70 complex (*Figure 9A–B*). As the unbound
drSSB was washed out from the reaction channel during drRecO mutant injection, drSSB appeared
to be tethered on dT70 in the intermediate state to recover to E ~ 0.6 of the drSSB–dT70 complex
(*Figure 9A–B*). drRecO mutants should also be tethered on ssDNA to maintain the low-FRET state
(E ~ 0.2). Otherwise, the E values of drSSB–dT70 would have moved to E ~ 0.6 immediately. When
the drRecO mutants successfully removed drSSB, the E value reached ~0.8 in *Figure 9A–B*. These
results clearly demonstrate that the occurrence of the low E intermediate state in the time trace is
closely related to the binding capability of drRecO to ssDNA. The weak binding affinity of the
drRecO mutants to ssDNA increased the fraction of the intermediate state and reduced the drSSB
displacement ability. The intermediate state was generated by drRecO partially unwinding ssDNA
from the drSSB–ssDNA complex, which results in a slightly stretched conformation of ssDNA before
fully unwinding ssDNA from drSSB (*Figure 9A–B*).

Unlike *E. coli* SSB, no strong interactions exist between the drSSB C-terminus and drRecO in DR
(*Ryzhikov et al., 2011*). However, *Cheng et al. (2014)*, using a native PAGE assay, reported that the
121-arginine of drRecO can be a key residue for the drSSB–drRecO interaction. They observed the

formation of the drRecO–drSSB complex at 5–25 µM concentrations without ssDNA. When we used R121A drRecO, no intermediate state was observed (*Figure 9—figure supplement 1*). It is possible that the heterotrimer becomes unstable due to the weakened interactions between R121A drRecO and drSSB, which renders the intermediate state difficult to observe at our time resolution.

In summary, our results demonstrate that drRecO binds to ssDNA sequentially using its two binding sites, which gives drRecO the ability to displace drSSB from ssDNA even though the binding affinity of drRecO for ssDNA is two orders of magnitude smaller than that of drSSB.

## Discussion

Homologous recombination is an important process in the repair of damaged DNA and is found in most organisms from bacteria to humans. Although the overall mechanism of this process is identical among different species, each species has a distinct mechanism for the initial step in recombinational repair, that is the displacement of SSB from ssDNA to load the recombinase onto ssDNA (*Inoue et al., 2008*; *Inoue et al., 2011*). DR has the fastest and most accurate DSB repair in terms of repair capacity by HR and is expected to have an inherent superefficient mechanism to displace SSB in favor of the repair machinery.

In this work, we demonstrate that drSSB binds to exposed-ssDNA much faster than drRecO. drRecO subsequently removes drSSB from ssDNA without the help of ATPs, despite the 300-fold higher binding affinity between drSSB and ssDNA. The displacement is not affected by the presence of free drSSB in solution. Three factors may enable drRecO to displace drSSB from ssDNA without using ATPs. The first factor is the two-step binding of drRecO to ssDNA, which relies on the species-specific structure of drRecO. drRecO has two major DNA-binding sites that locate both ends of the protein (*Leiros et al., 2005*). One of them is well known as an OB fold located in the N-terminus, a highly conserved DNA binding site of RecO proteins. The other binding site, a positive ridge, is a species-specific DNA-binding domain around the Zn-finger domain in the C-terminus. Both positively charged regions cover large areas of the drRecO surface: the residues that are essential for drRecO binding with DNA run diagonally on the upper side of the protein (the C-terminal positive ridge), and the N-terminal positively charged residues are in the lower half of the protein (*Figure 6—figure supplement 2C*). It is possible that ssDNA wraps drRecO through the positively charged regions, forming a helical shape with this structure (*Figure 6—figure supplement 2D*). Using one of the two binding sites, drRecO is able to form a heterotrimer complex with drSSB–ssDNA as an intermediate, and then disengages drSSB from the heterotrimer using the second binding site to finally form a drRecO–ssDNA complex (*Figure 9D*).

The second factor is the fast diffusion of drSSB on ssDNA. Ha and coworkers reported that the diffusion of SSB on ssDNA (the back-and-forth movement of SSB on ssDNA) is a critical factor for loading other proteins related to DNA metabolism onto SSB-coated ssDNA, and that the diffusion of SSB can be hindered when the length of the ssDNA is shorter than the occluded site size of SSB in *E. coli* (*Roy et al., 2009*). In our previous work, we showed that drSSB diffuses approximately ten times faster than ecoSSB on ssDNA (*Kim et al., 2015*). This should provide more opportunities to expose ssDNA transiently, allowing drRecO to bind the end of the ssDNA (*Figure 9D*). In this work, 40-nt ssDNA, which had a shorter length than the occluded site size of drSSB, showed much more frequent FRET transition events than longer lengths of ssDNA (*Figure 8*) and more easily re-bound to drSSB in the intermediate state upon failing to make a successful interaction with drRecO. These results suggest that drRecO has difficulty displacing drSSB from a shorter ssDNA because the diffusion of drSSB plays an important role in the displacement process. Although drRecO binds one end of ssDNA, drSSB removes drRecO from ssDNA as it rolls back to the original position on ssDNA. However, drSSB may diffuse to the opposite end of ssDNA because of the physical presence of drRecO on ssDNA. As drSSB diffuses to the opposite side of ssDNA, drRecO uses its second binding site to bind the other end of the ssDNA, and drSSB completely dissociates from ssDNA. As a result, drRecO facilitates RecA loading onto drSSB-coated ssDNA even without using ATP. The fast RecA loading without consuming ATP may help DR to repair damaged DNA for survival in extreme DNA-damaging environments.

The third factor is the difference in the occluded site sizes of drSSB and drRecO for the initial binding to ssDNA. Ha and coworkers suggested that continuous diffusion over tens of nucleotides would be important for protecting small DNA gaps and allowing access to other proteins

(*Zhou et al., 2011*). In competition, drRecO would have a better chance to bind to the small ssDNA gaps transiently exposed between diffusing drSSBs than unbound drSSB does, because drRecO can bind to a smaller region of free nucleotides than drSSB.

The process of SSB displacement by RecO is similar to the concentration-driven exchange mechanism (*Duderstadt et al., 2016*; *Åberg et al., 2016*; *Lewis et al., 2017*; *Spenkelink et al., 2019*) that arises from ssDNA-binding proteins that have multiple DNA-binding sites. In this model, partial dissociation of one side of the protein occurs without complete dissociation from ssDNA. Other ssDNA-binding proteins in solution then capture the nucleotides of ssDNA that were released by the resident protein, which accelerates the dissociation of the resident protein. In the case of drSSB–drRecO, the diffusion of drSSB may release nucleotides for the binding of drRecO, which allows the dissociation of drSSB. This process is comparable to the concentration-driven exchange mechanism. However, the reverse reaction, that is the dissociation of drRecO by a high concentration of drSSB, is different. Unlike the concentration-driven exchange, in which the concentration of each protein determines the direction of the exchange reaction, a high concentration drSSB is not able to dissociate drRecO from dT70 ssDNA (*Figure 3I*). It is possible that drSSB may fail to displace the resident drRecO from dT70 because drSSB requires a higher number of free nucleotides for ssDNA binding than does drRecO, and the movement of drRecO may not be active as drSSB on ssDNA (*Figure 3—figure supplement 1*). Two binding sites of drRecO may hinder its diffusion on ssDNA. Thus, the probability that drSSB binds enough ssDNA nucleotides would be much lower than that for drRecO. Although the drSSB–ssDNA complex is more stable than the drRecO–ssDNA complex thermodynamically, the formation of drRecO–ssDNA seems to be more favorable compared to that of drSSB–ssDNA kinetically (*Figure 9D*). As a result, concentration is not the only factor determining the direction of the exchange reaction in the drSSB–drRecO system, which is different from the concentration-driven exchange reaction.

In this work, we showed that drSSB binds to ssDNA much faster than drRecO does. Thus, when ssDNA is exposed, drSSBs will cover the ssDNA. drRecO subsequently removes drSSB from ssDNA, but the reverse reaction is not favored. The binding of drRecO to drSSB-coated ssDNA is relatively slow. Thus, a high concentration of drRecO may be required for the efficient removal of drSSB from ssDNA. Indeed, it has been reported that when DR is exposed to DNA damage, the expression level of drRecO significantly increases (*Joe et al., 2011*; *Beaume et al., 2013*). Thus, the reaction that we observed in vitro may occur in vivo.

Early genomic analysis reported that the genes for the RecFOR pathway are found more frequently than *recBCD* in bacterial genomes (*Rocha et al., 2005*). Among the genes of RecFOR, only 30% of bacterial genomes contain *recF*, whereas *recO* and *recR* were found in nearly 95% and 85% of bacterial genomes, respectively (*Garcia-Gonzalez et al., 2013*). Thus, it is possible that RecO together with RecR plays more important roles in the RecA-loading step for HR than does RecBCD through the RecOR pathway (*Sakai and Cox, 2009*). Our work shows how RecO removes SSB from ssDNA without the help of other recombination mediator proteins or ATP. Thus, bacteria that lack both *recBCD* and *recF* may use RecO to remove SSB from ssDNA, a process that might occur widely for DSB repair in bacteria.

# Materials and methods

**Key resources table**

| Reagent type (species) or resource | Designation | Source or reference | Identifiers | Additional information |
|---|---|---|---|---|
| Strain, strain background (*Escherichia coli*) | BL21(DE3) | Invitrogen | | Competent cells |
| Genetic reagent (*D. radiodurans*) | RecO | DOI: 10.2210/pdb1U5K/pdb | | |
| Commercial assay kit | PrimeSTAR DNA polymerase | Takara, Japan | | DNA polymerization for PCR |

*Continued on next page*

*Continued*

| Reagent type (species) or resource | Designation | Source or reference | Identifiers | Additional information |
|---|---|---|---|---|
| Software, algorithm | MATLAB | Mathworks Matlab Version R2018 | | The source code is distributed by T.J Ha group (http://bio.physics.Illinois.edu/) |
| Sequence-based reagent | DNA oligos | DNA oligos were ordered from IDT | | |

## Preparation of proteins and oligonucleotides

The *recO* gene was cloned into the pET21a vector from pDRO5 (the kind gift of Dr Sergey Korolev) in the N-terminal His-tagged form (pNK-RecO). The plasmid was transformed into *E. coli* strain BL21 (DE3) (Invitrogen), which was grown in LB broth in the presence of 50 µg/ml carbenicillin at 37°C to an optical density at 600 nm of 0.6. Cells were harvested by centrifugation and resuspended in lysis buffer containing 20 mM Tris-HCl (pH 7.4), 400 mM NaCl, 20 mM imidazole, 5 mM DTT and 2 mM AEBSF (4-[2-aminomethyl] benzenesulfonyl fluoride hydrochloride). After sonication on ice, cell debris and insoluble material were removed by centrifugation, and the protein was purified using a Ni-NTA column. The His-tagged protein was eluted with a buffer (15% glycerol [v/v], 20 mM Tris-HCl [pH 7.4], 100 mM NaCl, and 1 mM DTT) containing 500 mM imidazole and then dialyzed into storage buffer (15% glycerol [v/v], 20 mM Tris-HCl [pH 7.4], 100 mM NaCl, 1 mM DTT). The protein was divided into aliquots and stored at −80°C. The drSSB and drRecO mutants were expressed and purified using the same method described above.

K35E/R392, R195E/R196E drRecO mutants and drSSB (E120C) for fluorescent labeling were constructed by point mutations using PrimeSTAR DNA polymerase (TAKARA, Japan). Mutated plasmids were confirmed by sequence analysis (SOLGENT, Korea). drSSB (E120C) was labeled with maleimide-reactive Cy5 (GE healthcare, USA) according to the manufacturer's instructions and purified using a Sephadex-21 size exclusion column (GE Healthcare, USA).

DNA oligonucleotides were purchased from Integrated DNA Technologies (IDT), where 3AmMO and 5AmMC6 represent amine-modified thymine for fluorescent labeling, and 3Bio represents biotin modification for surface immobilization.

i.   /5AmMC6/TCG CTG CCG ACT CGA GAT CT/3Bio/
ii.  AGA TCT CGA GTC GGC AGC GA (T)n/3AmMO/, where n = 20, 30, 40, 50, 60, and 70.

Both DNA oligonucleotides, (i) and (ii), were labeled with the NHS ester-reactive Cy3B and Cy5 (GE Healthcare, USA), respectively, according to the manufacturer's instructions. We carried out ethanol precipitation to remove unreacted free dyes and annealed (i) and (ii) by mixing 500 nM of each in 10 mM Tris-HCl (pH 8.0) and 250 mM NaCl, followed by slow cooling from 90°C to room temperature.

## ALEX-FRET measurement

A schematic description of the ALEX microscope setup is presented in *Figure 1—figure supplement 1*. The ALEX setup that we used has been described extensively before (*Kapanidis et al., 2004*; *Lee et al., 2005*). To measure the FRET signal and the stoichiometry values between Cy3B and Cy5, we excited our samples with two lasers, a 532-nm solid-state green laser (Cobolt Samba, Cobolt AB) and a 633-nm He-Ne laser (25-LHP-925, Melles-Griot). The lasers were alternated using acoustic-optic modulators (23080–1, Neos Technologies) with a period of 100 µs, coupled by a dichroic mirror (z532bcm, Chroma). The lights were focused at 20 µm from the surface of a coverslip by a water-immersion objective (60×, 1.2 NA, Olympus) equipped in an inverted microscope (IX71, Olympus). Fluorescence emissions were collected through the objective, passed through a 100 µm pinhole and then separated by a beam-splitter (625DCLP, Chroma) and filtered by each signal, HQ580/60 m for Cy3b and HQ660LP for Cy5. The emissions were focused onto silicon avalanche photodiode detectors (APDs) (SPCM AQR-13, EG and G PerkinElmer) and measured for 10 min with a 0.6 ms bin time. All data were analyzed using home-built LABVIEW software (National Instruments).

To obtain the 2D E-S histogram, three different types of photons were analyzed from single-molecule bursts that were obtained by ALEX: a fluorescent emission of cy3B excited by the 532 -nm laser

($I_D^D$), a fluorescent emission of Cy5 excited by 532-nm laser ($I_D^A$), and a fluorescent emission of Cy5 excited by a 633-nm laser ($I_A^A$)). The E and S values were calculated by the following equations:

$$E = \frac{I_D^A}{I_D^D + I_D^A}, S = \frac{I_D^D + I_D^A}{I_D^D + I_D^A + I_A^A}$$

S is the stoichiometric value, which was used as a sorting parameter. For example, S ~ 1 implies that no acceptor dye (donor-only species) is detected in a single molecule because $I_A^A$ is 0. S ~ 0 implies that there is no donor dye (acceptor-only species) because $I_D^D + I_D^A$ = 0. The bursts for which 0.25<S<0.75 contain both donor and acceptor dyes. Thus, ssDNAs labeled with both donor and acceptor dyes were selected from the 2D E-S graph using the selection criterion 0.25<S<0.75 (boxes in 2D E-S graphs in *Figure 1B-E*).

To ensure the detection of single-diffusing ssDNA, we diluted samples to 50–100 pM (*Kim et al., 2012*; *Roy et al., 2008*). In *Figures 1C, E and F*, *2* µM drSSB was incubated with 1 nM dT70 for 10 min to ensure that all dT70 molecules bind to drSSB, and then the reaction mixture was diluted by adding 10-fold buffer solution for ALEX-FRET measurement. The final concentrations of drSSB and ssDNA were 200 nM and 100 pM, respectively. To observe the drSSB displacement by drRecO at the single molecule level, the mixture of 2 µM drSSB and 1 nM dT70 was diluted with a buffer containing 200 nM (*Figure 1G*) or 100 nM drRecO (*Figure 1H*). The final concentrations of drSSB and dT70 were 200 nM and 100 pM, whereas that of drRecO was 200 nM or 100 nM, respectively. The buffer for the single-molecule imaging contained 20 mM Tris-HCl (pH 7.4), 400 mM NaCl, 5% glycerol (v/v), 1 mM DTT, 100 µg/mL BSA and 1 mM MEA.

## Measurement of the apparent dissociation constants (K$_d$) of drSSB–ssDNA and drRecO–ssDNA by ALEX-FRET

The dissociation constants (K$_d$ in *Figure 1I*) of drSSB–ssDNA and drRecO–ssDNA were determined using ALEX-FRET to compare the strength of the binding interactions between the drSSB–ssDNA complex and the drRecO–ssDNA complex (*Kim et al., 2012*). The subpopulations of the drSSB–ssDNA complex and ssDNA were obtained from the 2D E-S graph (*Figure 1C*). Thus, we applied various concentrations of drSSB to 50 pM 70-nt ssDNA (dT70) (*Figure 1I*, left panel). The subpopulation at E = 0.09 corresponds to ssDNA, while the subpopulation at E = 0.58 corresponds to the drSSB–ssDNA complex. The concentration-dependent binding fraction ($\theta$), defined as $\theta$ = A$_1$/[A$_1$+A$_2$] (where A$_1$ and A$_2$ are the areas of the drSSB–ssDNA complex and of ssDNA, respectively), was fitted with the Hill equation yielding the K$_d$ values. We performed the same procedures using drRecO (*Figure 1I*, right panel).

## TIRF (total internal reflection fluorescence) microscopy

We used a home-built prism-type TIRF for smFRET and colocalization assays. The experimental setup for TIRF microscopy was the same as described before (*Roy et al., 2008*). A 532-nm green laser (Cobolt Samba, Cobolt AB) and a 633-nm He-Ne laser (25-LHP-925, Melles-Griot) were used to excite fluorophores. Total internal reflection was obtained on the surface of the flow channel by adjusting the incident angle of the excitation lasers. Photons emitted from the fluorophores were collected by a water-immersion objective lens (N.A. = 1.2, 60x oil, Olympus) in an inverted microscope (IX71, Olympus) and divided by a dichroic mirror (FF650-Di01−25 × 36, Semrock) built in a dual viewer (DV2, Photometrics). The fluorescence signals of the donor and acceptor dyes were filtered by FF01-S13580/40 (Semrock) and FF01-685/40 (Semrock), respectively, and detected by an EMCCD camera (iXon DU-897, Andor Technology). The fluorescence signals were recorded with 100 ms time resolution and analyzed by software distributed by the T.J. Ha group (http://bio.physics.Illinois.edu/). The quartz slide and coverslip were coated with a mixture of PEG and ~2% biotinylated PEG (Nektar Therapeutics) (*Rasnik et al., 2004*), and a flow chamber with an inlet and outlet was used for solution exchange. DNA was immobilized on the glass surface using biotin–NeutrAvidin interaction. All experiments were performed at room temperature in buffer (20 mM Tris-HCl, 400 mM NaCl, 0.1 mg/ml bovine serum albumin, 1 mg/ml glucose oxidase, 0.04 mg/ml catalase, and 0.4% [w/v] β-D-glucose) and 2 mM Trolox. The FRET values (E) were calculated using the aforementioned equation. We selected smFRET traces on the basis of the E values, obtained from ALEX-FRET, that were minimally affected by dye bleaching owing to short excitation time. To exclude dye

bleaching events on the binding fraction counts, we selected smFRET traces maintained over 3.5 min, which is enough to observe the RecO binding with a fairly slow association rate.

## Acknowledgements

The authors thank Jiwoong Kwon, Keewon Sung, Soyoung Bak, and Hye Ran Ko for discussion. This work was supported by grants from the Midcareer Research Program (NRF-2018R1A2B2001422 to SKK, NRF-2017R1A2B3010309 and NRF-2019R1A2C2090896 to NKL) of the National Research Foundation of Korea.

## Additional information

### Funding

| Funder | Grant reference number | Author |
|---|---|---|
| National Research Foundation of Korea | NRF-2017R1A2B3010309 | Nam Ki Lee |
| National Research Foundation of Korea | NRF-2018R1A2B2001422 | Seong Keun Kim |
| National Research Foundation of Korea | NRF-2019R1A2C2090896 | Nam Ki Lee |

The funders had no role in study design, data collection and interpretation, or the decision to submit the work for publication.

### Author contributions

Jihee Hwang, Conceptualization, Data curation, Formal analysis, Validation, Investigation; Jae-Yeol Kim, Cheolhee Kim, Conceptualization, Data curation, Investigation; Soojin Park, Data curation, Validation, Investigation; Sungmin Joo, Data curation, Investigation; Seong Keun Kim, Conceptualization, Supervision, Funding acquisition, Project administration; Nam Ki Lee, Conceptualization, Data curation, Supervision, Funding acquisition, Project administration

### Author ORCIDs

Nam Ki Lee ⓘD https://orcid.org/0000-0002-6597-555X

### Decision letter and Author response

Decision letter https://doi.org/10.7554/eLife.50945.sa1
Author response https://doi.org/10.7554/eLife.50945.sa2

## Additional files

### Supplementary files

• Transparent reporting form

### Data availability

All data generated or analysed during this study are included in the manuscript and supporting files. Source data files (459GB video images) are available upon request.

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
