## [Decision Letter]

**Acceptance summary:**

Your work presents single-molecule FRET studies to probe the mechanism of SSB removal by RecO. It provides a two-step mechanism to bring new perspectives into RecO-mediated DNA repair and will be of interest to the broad audience of *eLife*.

**Decision letter after peer review:**

Thank you for submitting your article "Single-molecule observation of a mechanically-driven SSB displacement by RecO" for consideration by *eLife*. Your article has been reviewed by three peer reviewers, one of whom is a member of our Board of Reviewing Editors, and the evaluation has been overseen by Cynthia Wolberger as the Senior Editor. The following individual involved in review of your submission has agreed to reveal their identity: Andrew Robinson (Reviewer #2).

The reviewers have discussed the reviews with one another and the Reviewing Editor has drafted this decision to help you prepare a revised submission.

Summary:

This manuscript by Hwang et al. used single-molecule FRET methods to study the mechanism of SSB removal by RecO. Because RecO loading on SSB-coated single stranded DNA is a key step in the repair of double strand break, the work is of potential interest to the broad audience of *eLife*. The authors found that RecO can efficiently displace stably bound SSB from ssDNA in 5 min without ATP, even when the binding of SSB to ssDNA is much tighter than RecO. Using cleverly designed RecO mutants and ssDNA substrates of different lengths, the authors showed that the exchange reaction like goes through a two-step mechanism, in which RecO first forms a heterotrimer intermediate with ssDNA and SSB using one of its two ssDNA binding sites, and then displaces SSB completely from ssDNA by wrapping around the ssDNA using its second binding site. In general, the work is clearly presented with impressive experimental design, execution and analysis, and will likely bring new mechanistic insight into RecO-mediated DNA repair.

Essential revisions:

1) The proposed mechanism of RecO-loading on SSB-coated ssDNA is contingent upon the presence of a critical heterotrimer intermediate ssDNA-SSB-RecO. The authors have provided convincing evidence showing that the low FRET state (intermediate state) became more frequently when shorter ssDNA or RecO mutant was used, and that this state was not from the transitory dissociation of SSB from ssDNA because SSB binding to ssDNA is very stable. Nevertheless, the reviewers would appreciate the authors to provide direct evidence demonstrating the presence/absence of the intermediate, such as a double-labeling experiment of both RecO and SSB, or adding RecO and SSB simultaneously to ssDNA in a competition experiment, which should have more of the intermediate state. Alternatively, the authors could address whether there is direction interaction between SSB and RecO on ssDNA, and whether such a direct interaction is important for the exchange reaction.

2) In order to determine the lifetime of SSB-ssDNA, the authors added excess unlabeled ssDNA so that once SSB dissociates it cannot bind to the FRET-labeled DNA and obtained a lifetime value close to an hour. However, just like RecO actively displacing SSB using multivalent binding, a new ssDNA could actively displace the resident ssDNA from the SSB molecule other than trapping spontaneously dissociated SSB passively. Therefore the true lifetime of SSB-ssDNA in the absence of excess ssDNA would be much longer. Determining the true dissociation constant of SSB and ssDNA is important for understanding the energetics of the proposed exchange mechanism. In fact, the authors can easily test this binding tethering ssDNA, adding SSB, rinsing free SSB away, and adding excess ssDNA. The high FRET species should return to the low FRET state rather quickly. If instead they stop after rinsing free SSB away, the high FRET species should remain for many hours or longer instead of just under an hour.

3) The authors need to formally test the possibility that SSB may also remove a resident RecO. What they see from such an experiment would be critical in coming up with a quantitative model. The current set of data do not clarify how RecO can easily remove SSB. One guess is that for initial loading of RecO requires much fewer numbers of free nucleotides than SSB so that the reaction in the other direction is not favored.

4) It appeared that all the experiments was conducted by adding excess amount of RecO to preformed ssDNA-SSB complex, but in several places it was not entirely clear. It is important to know whether the surface-based experiments have SSB present in solution as RecO (or its mutants) are introduced, or whether unbound SSB is washed out when RecO is injected. The authors' major conclusions and the interest of the field will be contingent upon sufficient details provided in the manuscript.

For example, the data in Figure 1 was collected with SSB present in solution at the same time as RecO. The solution FRET data displayed in Figure 1 suggest that RecO is displacing SSB from the ssDNA even when SSB is present in solution, although the only evidence of this is indirect – the transition to a 0.78 FRET state. In the surface measurements, RecO is clearly displacing labelled SSB from the ssDNA, but here there is no SSB in solution. The model that the authors proposed in Figure 8 provides a tangible explanation for replacement of SSB by RecO in the absence of SSB in solution, although it is important to note that this is essentially the same mechanism put forward by the van Oijen lab to explain concentration-driven exchange of proteins within the replisome. If RecO is able to chase SSB off the DNA while SSB remains in solution, in the absence of an external energy source, that would be truly remarkable given that SSB binds more tightly to ssDNA than RecO. It's hard to imagine how such a mechanism could work.

Additionally, does the condition of excess amount of RecO mimic what's happening in vivo? According to the in vitro measurements, RecO binds more slowly and weaker to ssDNA than SSB. Therefore, in order to displace SSB effectively, high concentration of RecO to compete off SSB is critical. In vivo the situation could be different and that SSB and RecO may compete for binding to ssDNA simultaneously. It would be helpful if the authors discuss such a scenario affect RecO's function in vivo.

5) The exchange mechanism is quite similar to a concentration-driven exchange mechanism described by the van Oijen lab (see Duderstadt et al., 2016; Åberg et al., 2016; Lewis et al., 2017; and see in particular Spenkelink et al., 2019, which concerns exchange reactions of the *E. coli* SSB protein). The van Oijen studies have focused on homo-exchanges; exchange of one SSB tetramer for another SSB tetramer, for example. The current study describes a hetero-exchange, in which one ssDNA-binding protein with multiple DNA-binding sites (SSB) is exchanged with another ssDNA-binding protein with multiple DNA-binding sites (RecO). The current work should discuss the similarity of the proposed mechanism with the ones mentioned above.

6) The authors need to interpret their results in the context of what actually takes place in cells v.s. what happens in vitro. First, the RecR protein is not present in the in vitro condition. In *Deinococcus* and other organisms, RecO operates as a complex with RecR. Second, the RecA protein is not present. Third, SSBs from other organisms bind ssDNA with a high degree of cooperativity. The short ssDNAs used in the study would presumably only accommodate a single SSB tetramer, thus cooperative SSB binding would not be possible. Fourth, although it appears that SSB is probably binding ssDNA in a fully wrapped mode (akin to the 65-mode of ecSSB) under these conditions, but this has not been determined. As their proposed mechanism relies on SSB binding in a fully wrapped mode, it is important to rule out the possibility that it is binding in a partially wrapped state (akin to the ecSSB 35-mode).

7) The authors attribute each of the models displayed in Figure 3 to behaviors associated with particular organisms. It is important to appreciate that each of the studies that the authors refer to had completely different experimental designs, each with its own limitations. According to Inouye et al. (to whom the authors refer in the context of Model 2), the state illustrated as Model 3 is actually an intermediate formed en route to the state pictured as Model 2. While they propose that the state shown as Model 2 exists, they do not claim that it is the end of the reaction. It seems likely that the states shown as Model 3 and Model 2 are simply intermediates towards the state shown in Model 1.

8) The observation that ATP is unnecessary is interesting. At the single molecule level, energy will be needed to drive the displacement of SSB from ssDNA, especially that the Kd of ssDNA with SSB is orders of magnitude higher than that with RecO. Can they authors discuss what energy source could be used to drive this process? Additionally, the word "mechanically-driven" in the title may not be appropriate. It appeared to refer to that there is no chemical energy such as ATP hydrolysis involved, but other chemical energy such as binding energies (most likely) are involved to drive the exchange reaction.

[Editors' note: further revisions were suggested prior to acceptance, as described below.]

Thank you for resubmitting your work entitled "Single-molecule observation of ATP-independent SSB displacement by RecO in *Deinococcus radiodurans*" for further consideration by *eLife*. Your revised article has been evaluated by Cynthia Wolberger as the Senior Editor, and a Reviewing Editor.

The manuscript has been improved but there are some remaining issues that need to be addressed before acceptance, as outlined below: (1) a more in-depth discussion of the physical basis for the irreversibility of the model in Figure 9, and (2) careful textual editing of the article. Please see specific comments of the reviewers below.

Reviewer #2:

This manuscript by Hwang et al. is a revised version of an earlier manuscript. The authors have done an excellent job in addressing concerns around experimental design and the description results that were raised in the first round of review. The authors now clearly demonstrate that drRecO can exchange for drSSB on ssDNA (at least for DNAs up to 70 nt), and that the exchange is non-reversible (at least on the timescale of the experiment). The authors have outlined factors that are likely to contribute to the mechanism of this process: rapid diffusion of drSSB, two-step binding of drRecO, smaller binding footprint of drRecO, etc. Exactly how these factors lead to the non-reversibility of the exchange reaction remains unclear to me. In Figure 9D, the model is outlined in a series of five steps. Steps 1 and 2 are depicted as reversible, while steps 3 – 5 are depicted as non-reversible. This depiction is consistent with the experimental data, however I do not understand what the physical basis for the non-reversibility is. I cannot understand it from a thermodynamics perspective. I could perhaps imagine a model driven by kinetics.

The current study provides a solid set of observations. Studies into the physical basis of the phenomenon will surely follow. I would like to see the work published as it stands (although I would recommend some careful editing throughout to correct grammar errors prior to publication).

Reviewer #3:

I have reviewed the authors' responses and revised manuscript carefully. They have performed a lot of new experiments that addressed all of the major concerns, and their analysis and conclusions are on a firm ground. I am satisfied with the revision and recommend publication.

---

## [Author Response]

Essential revisions:1) The proposed mechanism of RecO-loading on SSB-coated ssDNA is contingent upon the presence of a critical heterotrimer intermediate ssDNA-SSB-RecO. The authors have provided convincing evidence showing that the low FRET state (intermediate state) became more frequently when shorter ssDNA or RecO mutant was used, and that this state was not from the transitory dissociation of SSB from ssDNA because SSB binding to ssDNA is very stable. Nevertheless, the reviewers would appreciate the authors to provide direct evidence demonstrating the presence/absence of the intermediate, such as a double-labeling experiment of both RecO and SSB, or adding RecO and SSB simultaneously to ssDNA in a competition experiment, which should have more of the intermediate state. Alternatively, the authors could address whether there is direction interaction between SSB and RecO on ssDNA, and whether such a direct interaction is important for the exchange reaction.

We appreciate the valuable comments of the reviewers. We performed all three experiments, as suggested by the reviewers.

The direct method to observe the hetero-trimer is dual-labeling of drSSB and drRecO. Thus, we have considered dual-labeling of drSSB and drRecO to prove our hypothesis of the hetero-trimer. However, the replacement of drSSB by drRecO typically requires >10 nM drRecO to detect drSSB-displacement with a statistically meaningful number of events in our observation time window. Single-molecule detection of drRecO should be difficult at such a high concentration. Thus, we have proved that the intermediate state is the drRecO-ssDNA-drSSB trimer by showing the dynamics of drSSB replacement by drRecO using various lengths of ssDNAs and drRecO mutants. Nevertheless, we performed the direct-labeling experiment, as suggested by the reviewer. A single point mutation was introduced in drRecO (two mutants of A84C and A209C) and labeled with a fluorophore (Cy3 or Cy5). The drSSB mutant (G120C) that is shown in Figure 4 was used. As expected, we could not detect single molecule signal from dye-labeled drRecO in >10 nM concentrations due to the high background level. Thus, a dual-labeling scheme could not be employed in this study.

Next, to determine whether the intermediate state more frequently appears under competitive binding conditions of drRecO and drSSB, we simultaneously added and allowed drRecO and drSSB to competitively bind to ssDNA. Figure 3E presents the experimental scheme and FRET time trace. The dotted orange line in the time-trace represents the time point that drRecO and drSSB were injected together. After the addition of drSSB and drRecO to the surface-tethered dT70, the E value increased immediately from ~0.2 to ~0.6, which was followed by the second increment of the E value from ~0.6 to ~0.8. This indicates that drSSB immediately binds to dT70 and thus forms the same initial state of the pre-incubated drSSB-ssDNA complex. This result is consistent with the measurement that the association rate of drSSB is approximately 10,000 times larger than that of drRecO [Witte et al., 2005]. Thus, the effect of simultaneous addition of both drRecO and drSSB does not differ from that of using preformed drSSB-dT70. Indeed, the fraction that shows the intermediate state remained <2%, which is comparable to the preformed experiment. Overall, due to the significant differences in the association rates, the competition experiment results the same findings as using the pre-formed drSSB-ssDNA complex.

Next, we tested whether the unbound drSSB could be a competitor of drRecO for binding to drSSB-coated ssDNA (Figure 3F-H). For the purpose of comparison, we newly performed drSSB-displacement by drRecO using pre-formed drSSB-dT70 in Figure 3A-D. It is to be noted that drSSB was washed out in the pre-formed drSSB-dT70 experiment, as shown in Figure 3A-D. When 0.2 μM drRecO was added in the presence of 0 μM, 0.2 μM, and 0.5 μM drSSB in solution (Figure 3B, F, and G, respectively), the exchange rates were nearly invariant. However, the exchange rates increased as the concentration of drRecO increased from 0.2 μM to 0.5 μM in the presence of 0.5 μM drSSB (Figure 3G and H). The exchange rate was independent of the drSSB concentration in solution. These results indicate that unbound drSSB in solution is not a competitor of drRecO for drSSB-coated ssDNA binding. We further report the effects of unbound drSSB in solution in our response to Comment #3. We have added these results in Figure 3A-H of the revised manuscript and added text in the subsection “Real-time observation of drSSB displacement by drRecO using single-molecule FRET”.

Lastly, we investigated the effect of direct interactions between drRecO and drSSB. Unlike *E. coli* SSB, no strong interaction exists between drSSB C-terminus and drRecO in DR [Ryzhikov et al., 2011]. However, Cheng et al., using a native PAGE assay, reported that the 121-arginine of drRecO can be a key residue for the drSSB-drRecO interaction [Cheng et al., 2014]. They observed formation of the drRecO-drSSB complex in 5 – 25 µM concentrations without ssDNA. To investigate the role of direct interactions between drRecO-drSSB in the intermediate state, we constructed a R121A drRecO mutant (Figure 9—figure supplement 1A) and evaluated its binding properties to ssDNA (Figure 9—figure supplement 1). R121A drRecO showed weak ssDNA binding ability in the cumulative FRET histogram (Figure 9—figure supplement 1B), which is comparable to that of the C-/N-terminal mutants (Figure 7). This may be due to the position of the 121-arginine, which is located at the middle of the expected ssDNA wrapping structure (Figure 6—figure supplement 2) and blocks the full helical binding of ssDNA. A new peak around E = 0.6 was observed for R121A drRecO, which was not observed for the K35E/R39E and R195E/R196E drRecO mutants. This mid-FRET state may correspond to a half-ssDNA bound drRecO resulted by failed two-step binding. An important feature of R121A is the absence of the intermediate state during drSSB displacement, unlike with other mutants (Figure 9—figure supplement 1C). When R121A drRecO was added to the preformed drSSB-dT70 complex, R121A drRecO exhibited similar efficiency for drSSB displacement with R195E/R196E drRecO (Figure 9—figure supplement 1D), which is consistent with its ssDNA binding capability. We have shown that, when drSSB-displacement occurred by R195E/R196E drRecO, 51% of the time-traces exhibited the intermediate state (Figure 9—figure supplement 1D). However, when R121A drRecO was used, no intermediate state was observed for >200 time-traces. It is highly possible that the hetero-trimer becomes unstable due to the weakened interactions between R121A drRecO and drSSB, which renders the intermediate state difficult to observe given our time resolution. These results suggest that the weak direct interactions between drRecO and drSSB contribute to the formation of the intermediate state, which thus further supports our interpretation that the intermediate state is the hetero-trimer of drSSB-drRecO-ssDNA. However, these interactions must be weak to completely dissociate drSSB from the drRecO-ssDNA complex.

We added these results in Figure 9—figure supplement 1 and added text in the third paragraph of the subsection “drRecO displaces drSSB from ssDNA by using a sequential two-step binding mode without consuming ATP”.

2) In order to determine the lifetime of SSB-ssDNA, the authors added excess unlabeled ssDNA so that once SSB dissociates it cannot bind to the FRET-labeled DNA and obtained a lifetime value close to an hour. However, just like RecO actively displacing SSB using multivalent binding, a new ssDNA could actively displace the resident ssDNA from the SSB molecule other than trapping spontaneously dissociated SSB passively. Therefore the true lifetime of SSB-ssDNA in the absence of excess ssDNA would be much longer. Determining the true dissociation constant of SSB and ssDNA is important for understanding the energetics of the proposed exchange mechanism. In fact, the authors can easily test this binding tethering ssDNA, adding SSB, rinsing free SSB away, and adding excess ssDNA. The high FRET species should return to the low FRET state rather quickly. If instead they stop after rinsing free SSB away, the high FRET species should remain for many hours or longer instead of just under an hour.

We appreciate this comment and the excellent suggestion. We agree with the reviewer’s comments on the dissociation. According to the reviewer’s suggestion, we performed the surface-tethering experiment by employing a TIRF microscope to measure the dissociation rate of drSSB more accurately in the absence of the free dT70 (revised Figure 1—figure supplement 2). drSSB was incubated with the surface-tethered dT70 for 5 min and then the free drSSBs were washed out by injecting a buffer. We then measured the fraction of drSSB that was dissociated from dT70. To reduce the artifacts of photo-bleaching, we acquired the images for 40 sec at each time point and checked the non-bleaching of Cy5 by introducing an additional red laser (638 nm CW laser) for 10 sec. All data points in revised Figure 1—figure supplement 2 were acquired from >300 molecules. The fraction of drSSB that was dissociated from dT70 was only 5.2% for 2 hours. We sealed the channels and scavenged the dissolved oxygens to protect the fluorophore from damage induced by the oxygen due to long-term observation, but the bleaching and blinking of the fluorophores were drastically increased as the acquisition time proceeded over 2 hours. As the reviewer expected, we confirmed that drSSB binds more stably to ssDNA in the absence of free ssDNA compared to in the presence of free ssDNA. Thus, we modified the manuscript according to the result of revised Figure 1—figure supplement 2.

3) The authors need to formally test the possibility that SSB may also remove a resident RecO. What they see from such an experiment would be critical in coming up with a quantitative model. The current set of data do not clarify how RecO can easily remove SSB. One guess is that for initial loading of RecO requires much fewer numbers of free nucleotides than SSB so that the reaction in the other direction is not favored.

We appreciate this critical comment. Since SSB is the first protein that binds in a physiological environment to the exposed ssDNA during DNA metabolism, including the DNA repair process, we did not consider the reverse reaction, i.e., RecO-replacement by SSB, in this work. However, we fully agree with the reviewer’s opinion that further experimentation to confirm whether the reverse reaction occurs is required for quantitative modeling.

First, we investigated whether drSSB removes the resident drRecO from dT70 (Figure 3I). We incubated 500 nM drRecO with surface-tethered dT70s for 10 min, which allows drRecO to fully bind to dT70. The FRET value once again increased to ~0.8 by the binding of drRecO to dT70. We then injected drSSB to the drRecO-dT70 complex (free drRecO was washed out during drSSB injection). We observed that the FRET value was maintained in the presence of 0.5 μM drSSB (Figure 3I). This result indicates that drSSB cannot remove drRecO from the drRecO-dT70 complex.

Next, we performed additional experiments to understand why the reverse reaction was not favored. As the reviewer suggested, the number of free nucleotides may be an important factor. To check this possibility, we used ALEX-FRET to observe binding between the proteins and short ssDNAs and to measure how many nucleotides are required to load drSSB and drRecO onto ssDNA (Figure 3—figure supplement 1). Either 500 nM drSSB or 2 μM drRecO were incubated with ssDNAs (dT20, dT30 and dT40) for 30 min to construct fully bound complexes. We observed that drRecO changed the FRET values of all ssDNAs of dT20, dT30, and dT40, which indicates high and low FRET that correspond to fully bound and partially bound complexes. It is to be noted that drRecO does not bind to dT20, as shown Figure 5—figure supplement 1 of the original manuscript. In this figure, the incubation time was only 10 min, but when we increased the incubation time to 30 min, drRecO is found to partially bind to dT20. However, compared to drRecO, drSSB only changed the FRET value of dT40. Witte et al. reported that drSSB binds to ssDNA with 47-54 nt to form fully wrapped drSSB-ssDNA complex [Witte et al., 2005]. Thus, drSSB seemed to form a partially bound complex with dT40 and did not bind to dT20 and dT30. drRecO formed a nearly fully bound complex with dT30 and dT40 and a partially bound complex with dT20. Thus, drSSB may not displace the resident drRecO from the preformed drRecO-ssDNA complex because drSSB requires a high number of free nucleotides for ssDNA binding compared to drRecO.

By clarifying a reptation mechanism for SSB diffusion along ssDNA using optomechanical experiment, Ha and coworkers suggested that the continuous diffusion over tens of nucleotides would be important for protecting small DNA gaps and allowing access to other proteins [Zhou et al., 2011]. drRecO, compared to unbound drSSB, would be more efficient in binding to the transiently exposed small ssDNA gaps between diffusing drSSBs, as a smaller binding site size is required for drRecO in the active DNA repairing state of DR.

We included this result in Figure 3I and Figure 3—figure supplement 1 and discussed this result in the last paragraph of the subsection “Real-time observation of drSSB displacement by drRecO using single-molecule FRET”.

4) It appeared that all the experiments was conducted by adding excess amount of RecO to preformed ssDNA-SSB complex, but in several places it was not entirely clear. It is important to know whether the surface-based experiments have SSB present in solution as RecO (or its mutants) are introduced, or whether unbound SSB is washed out when RecO is injected. The authors' major conclusions and the interest of the field will be contingent upon sufficient details provided in the manuscript.For example, the data in Figure 1 was collected with SSB present in solution at the same time as RecO. The solution FRET data displayed in Figure 1 suggest that RecO is displacing SSB from the ssDNA even when SSB is present in solution, although the only evidence of this is indirect – the transition to a 0.78 FRET state. In the surface measurements, RecO is clearly displacing labelled SSB from the ssDNA, but here there is no SSB in solution. The model that the authors proposed in Figure 8 provides a tangible explanation for replacement of SSB by RecO in the absence of SSB in solution, although it is important to note that this is essentially the same mechanism put forward by the van Oijen lab to explain concentration-driven exchange of proteins within the replisome. If RecO is able to chase SSB off the DNA while SSB remains in solution, in the absence of an external energy source, that would be truly remarkable given that SSB binds more tightly to ssDNA than RecO. It's hard to imagine how such a mechanism could work.Additionally, does the condition of excess amount of RecO mimic what's happening in vivo? According to the in vitro measurements, RecO binds more slowly and weaker to ssDNA than SSB. Therefore, in order to displace SSB effectively, high concentration of RecO to compete off SSB is critical. In vivo the situation could be different and that SSB and RecO may compete for binding to ssDNA simultaneously. It would be helpful if the authors discuss such a scenario affect RecO's function in vivo.

We appreciate this important comment. As suggested by the reviewer’s comment, we provided more detailed information on the unbound drSSB of each experiment in the revised manuscript. In the surface-tethered experiments of the original manuscript, drSSB was washed out during drRecO injection. We added the following sentence: “During the injection of drRecO, unbound drSSB was washed out in the surface-tethered experiments.” “In the surface-tethered experiment, unbound drSSB was washed out during drRecO injection.” We also included the following sentence in the captions of Figure 2, Figure 4, Figure 8, and Figure 9, “During the injection of drRecO, unbound drSSB was washed out.”

Thanks to the reviewer’s comments, we newly showed in this revised manuscript that drRecO can displace drSSB in the presence of drSSB in solution. We performed drSSB-displacement by drRecO in the presence of 0 μM, 0.2 μM, and 0.5 μM drSSB in solution using surface-tethered measurements (Figure 3). The rate of drSSB-displacement by drRecO was nearly independent of drSSB, while the exchange rate increased as the concentration of drRecO increased (Figure 3). This result is further supported by the measurement of the reverse reaction, i.e., drRecO-displacement by drSSB in Figure 3I. After washing out drRecO in solution, we added 0.5 μM drSSB. Thus, drSSB could not displace drRecO from ssDNA. As the reviewer suggested in the comment #3, we investigated the effect of free nucleotides to understand why the reverse reaction was not favored (Figure 3—figure supplement 1). Here, we found that drRecO can bind to smaller nucleotides compared to drSSB. These results have been incorporated into the revised manuscript (subsection “Real-time observation of drSSB displacement by drRecO using single-molecule FRET”, last paragraph).

We appreciate the comment on the concentration-driven exchange. As for the concentration-driven exchange mechanism, this was discussed in our response to comment #5.

We also appreciate the comment on the in vivo. We showed that drSSB binds to ssDNA much faster than does drRecO. Thus, when ssDNA is exposed, drSSBs will cover ssDNA. drRecO subsequently removes drSSB from ssDNA, but the reverse reaction is not favored, as shown in Figure 3I. As the reviewer commented, the binding of drRecO to the drSSB-coated ssDNA is relatively slow. Thus, a high concentration of drRecO may be required for efficient removal of drSSB from ssDNA. Indeed, it has been reported that, when DR is exposed to DNA damage, the expression level of drRecO significantly increases [Joe et al., 2011; Beaume et al., 2013]. In these works, the expression levels of proteins that are related to DNA metabolism were quantified by detecting their transcriptosomes with qRT-PCR, as the dose of γ-radiation and chemical-induced DNA damage increased. The expression levels of drRecO increased up to 11.8-times and the increased level was higher than most of those of other proteins [Joe et al., 2011; Beaume et al., 2013]. Thus, the reaction that we observed in vitro is deemed to occur in vivo.

We compared our in vitro work and in vivo work in the “Discussion” section (fifth and sixth paragraphs).

5) The exchange mechanism is quite similar to a concentration-driven exchange mechanism described by the van Oijen lab (see Duderstadt et al., 2016l; Åberg et al., 2016; Lewis et al., 2017; and see in particular Spenkelink et al., 2019, which concerns exchange reactions of the *E. coli* SSB protein). The van Oijen studies have focused on homo-exchanges; exchange of one SSB tetramer for another SSB tetramer, for example. The current study describes a hetero-exchange, in which one ssDNA-binding protein with multiple DNA-binding sites (SSB) is exchanged with another ssDNA-binding protein with multiple DNA-binding sites (RecO). The current work should discuss the similarity of the proposed mechanism with the ones mentioned above.

We appreciate this important comment. As the reviewer commented, the concentration-driven exchange that was described by the van Oijen group arises from ssDNA-binding proteins that have multiple DNA-binding sites. In this model, a partial dissociation of one side of the protein occurs without complete dissociation from ssDNA. Other ssDNA-binding proteins in solution then capture the nucleotides of ssDNA that were released by the resident protein, which accelerates the dissociation of the resident protein.

The process of SSB displacement by RecO is similar to the concentration-driven exchange mechanism. In the case of the drSSB-drRecO system, the diffusion (moving on ssDNA without dissociation) of drSSB on ssDNA may release nucleotides for the binding of drRecO, which allows the dissociation of drSSB. This process is comparable to the concentration-driven exchange mechanism. However, the reverse reaction, i.e., the dissociation of drRecO by a high concentration of drSSB, is different. Unlike the concentration-driven exchange, in which the concentration of each protein determines the direction of the exchange reaction, a high concentration of drSSB was not able to dissociate drRecO from dT70 ssDNA. It is possible that drSSB could not displace the resident drRecO from dT70, because drSSB for the ssDNA binding requires a high number of free nucleotides compared to drRecO (Figure 3I and Figure 3—figure supplement 1). As drSSB diffuses on ssDNA (moving on ssDNA without dissociation), it has more chance to expose ssDNA for the binding of other proteins than drRecO does. As a result, the concentration is not the only determining factor for the direction of the exchange reaction in the drSSB-drRecO system, which is different from the concentration-driven exchange reaction. We clearly mentioned this in the “Discussion” section (fifth paragraph).

6) The authors need to interpret their results in the context of what actually takes place in cells v.s. what happens in vitro. First, the RecR protein is not present in the in vitro condition. In Deinococcus and other organisms, RecO operates as a complex with RecR. Second, the RecA protein is not present. Third, SSBs from other organisms bind ssDNA with a high degree of cooperativity. The short ssDNAs used in the study would presumably only accommodate a single SSB tetramer, thus cooperative SSB binding would not be possible. Fourth, although it appears that SSB is probably binding ssDNA in a fully wrapped mode (akin to the 65-mode of ecSSB) under these conditions, but this has not been determined. As their proposed mechanism relies on SSB binding in a fully wrapped mode, it is important to rule out the possibility that it is binding in a partially wrapped state (akin to the ecSSB 35-mode).

We appreciate this valuable comment. Indeed, RecO forms a complex with RecR in most bacterial species. However, each species has distinct features for the stoichiometry and structure of the RecOR complex and their functions are also diverse [Radzimanowski et al., 2013]. In *E. coli*, both ecoRecO and ecoRecR are required for ecoRecA-loading to ecoSSB-coated ssDNA [Umezu et al., 1994; Bell et al., 2012]. However, in *Thermos thermophiles,* ttRecO operates to displace ttSSB from ssDNA, and ttRecR then assists ttRecA loading to ssDNA [Inoue et al., 2008]. In *Bacillus subtilis,* bsRecO is essential and sufficient to load bsRecA onto bsSSB-coated ssDNA without bsRecR [Manfredi et al., 2008; Carrasco et al., 2015]. In DR, Radzimanowski J. et al. proposed a model for the assembly of the drRecOR complex on DNA by analyzing the structure of the drRecOR complex and its DNA binding mode and suggested that drRecO is involved in the binding and removal of SSB from DNA [Radzimanowski et al., 2013]. *recO* and *recR* were found in nearly 95% and 85% of bacterial genomes, respectively (Garcia-Gonzalez et al., 2013), which means that RecO does not always function with RecR. Our work suggests that drRecO is sufficient to dissociate drSSB from ssDNA. In our preliminary work, we observed that the addition of the drRecO-drRecR complex did not improve the efficiency of the drSSB displacement from ssDNA, which requires further investigation to obtain a definitive conclusion. In this study, we clarified the mechanism that underlies drSSB dissociation by drRecO without the aid of ATP for the efficient exchange in DR. Although the overall mechanism of the DNA repair process by HR is identical in most organisms, the detailed molecular mechanisms of the initial HR process differ across species. Thus, our study will be a mechanistic model for the ATP-independent SSB-displacement process by mediator proteins with multiple-DNA binding sites.

However, we agree that the present study does not incorporate all components for RecA-loading, which will require much further research. We currently focused on the function of drRecO and its unique mechanism of drSSB-displacement without the use of ATPs. As stated in this manuscript, this is the first study to show the function of individual proteins among the RecFOR pathway. We are currently investigating the function of the drRecOR complex with the aim to reveal the entire RecFOR pathway in the near future.

drSSB acts as a homo-dimer with two OB-folds per monomer, whereas ecoSSB acts as a homo-tetramer with one OB-fold per monomer. Moreover, ecoSSB shows a large difference in the occluded site size between binding modes depending on the salt concentration (35 nt at <200 mM NaCl and 65 nt at >200 mM NaCl). As for drSSB, the occluded site size is similar as 47 nt at <200 mM NaCl and 54 nt at >200 mM NaCl. Distinct features exist between species regarding SSB binding to ssDNA. In addition, as we mentioned in the “Discussion” section, additional species-specific features of drSSB, a rapid diffusion rate and lower ssDNA binding energy are factors that enable drRecO to replace SSB efficiently.

It has been previously reported that drRecO binds to ssDNA that is composed of poly dT with 70 nt in a fully wrapped mode using the binding-isotherm and stopped-flow kinetics experiments [Witte et al., 2005; Kozlov et al., 2010]. We used the experimental conditions to ensure the fully wrapping mode. In addition, drSSB is cooperative under low salt conditions, but not under high salt conditions [Witte et al., 2005; Kozlov et al., 2010]. Generally, the binding energy of drSSB at high salt concentrations is higher than that at low salt concentrations [Kozlov et al., 2010; Zhou et al., 2011]. In the present study, all experiments were performed in high salt conditions. Unfortunately, the FRET states of drSSB and drRecO were not distinguishable in low salt conditions.

7) The authors attribute each of the models displayed in Figure 3 to behaviors associated with particular organisms. It is important to appreciate that each of the studies that the authors refer to had completely different experimental designs, each with its own limitations. According to Inouye et al. (to whom the authors refer in the context of Model 2), the state illustrated as Model 3 is actually an intermediate formed en route to the state pictured as Model 2. While they propose that the state shown as Model 2 exists, they do not claim that it is the end of the reaction. It seems likely that the states shown as Model 3 and Model 2 are simply intermediates towards the state shown in Model 1.

We agree with the reviewer’s opinion that Model 2 and Model 3 could be intermediate states prior to full dissociation of SSB from ssDNA. Thus, we clearly mentioned that Model 2 and Model 3 could be intermediate states in the revised manuscript. We also toned down our classification of the models, as shown below.

“The most intuitive model is that RecO replaces and fully dissociates SSB from ssDNA (Model 1). However, in *Thermus thermophilus* (tt), which is phylogenetically related to DR, ttRecO displaces ttSSB by strongly binding to the DNA, but the unbound ttSSB remains tethered (Model 2) (Inoue et al., 2008; Inoue et al., 2011). […] It seems to be common in other bacteria that SSB remains in the complex of RecO-ssDNA as an intermediate or final state.”

8) The observation that ATP is unnecessary is interesting. At the single molecule level, energy will be needed to drive the displacement of SSB from ssDNA, especially that the Kd of ssDNA with SSB is orders of magnitude higher than that with RecO. Can they authors discuss what energy source could be used to drive this process? Additionally, the word "mechanically-driven" in the title may not be appropriate. It appeared to refer to that there is no chemical energy such as ATP hydrolysis involved, but other chemical energy such as binding energies (most likely) are involved to drive the exchange reaction.

We appreciate this comment. The reviewers also mentioned the concentration-driven exchange mechanism in comments #4 and #5. As we answered these questions in the revised manuscript, the mechanism of drSSB-displacement by drRecO is quite comparable to the concentration-driven exchange mechanism. However, unlike the concentration-driven exchange mechanism, in which the reaction direction is mostly determined by the concentration of reaction components, the reverse reaction, i.e., drRecO-displacement by drSSB, is not determined by the concentrations of drSSB and drRecO. This may be due to the different ssDNA-binding properties between drSSB and drRecO. We clearly mentioned this in the “Discussion” section fifth paragraph), following the reviewers’ comments.

We initially chose “mechanically-driven” with the aim to distinguish our work from the enzymatic processes of typical SSB displacement, which use chemical energy (i.e.ATP hydrolysis). Besides, the exchange reaction could not be explained by a simple comparison between the binding energies that were inferred from the K_d_ values of drSSB and drRecO. We showed that drRecO displaces drSSB from ssDNA using a sequential two-step binding mode without the aid of ATP. However, we agree that “mechanically-driven” may also be inappropriate. Thus, we changed the title of this manuscript to as follows: “Single-molecule observation of ATP-independent SSB displacement by RecO in *Deinococcus radiodurans*”.

[Editors' note: further revisions were suggested prior to acceptance, as described below.]

The manuscript has been improved but there are some remaining issues that need to be addressed before acceptance, as outlined below: (1) a more in-depth discussion of the physical basis for the irreversibility of the model in Figure 9, and (2) careful textual editing of the article. Please see specific comments of the reviewers below.Reviewer #2:This manuscript by Hwang et al. is a revised version of an earlier manuscript. The authors have done an excellent job in addressing concerns around experimental design and the description results that were raised in the first round of review. The authors now clearly demonstrate that drRecO can exchange for drSSB on ssDNA (at least for DNAs up to 70 nt), and that the exchange is non-reversible (at least on the timescale of the experiment). The authors have outlined factors that are likely to contribute to the mechanism of this process: rapid diffusion of drSSB, two-step binding of drRecO, smaller binding footprint of drRecO, etc. Exactly how these factors lead to the non-reversibility of the exchange reaction remains unclear to me. In Figure 9D, the model is outlined in a series of five steps. Steps 1 and 2 are depicted as reversible, while steps 3 – 5 are depicted as non-reversible. This depiction is consistent with the experimental data, however I do not understand what the physical basis for the non-reversibility is. I cannot understand it from a thermodynamics perspective. I could perhaps imagine a model driven by kinetics.The current study provides a solid set of observations. Studies into the physical basis of the phenomenon will surely follow. I would like to see the work published as it stands (although I would recommend some careful editing throughout to correct grammar errors prior to publication).

We appreciate the reviewer #2’s comments.

A) We agree that drRecO-ssDNA is less stable compared with drSSB-ssDNA thermodynamically. We mentioned three factors, which may enable drRecO compete with drSSB, in the manuscript. These factors are linked with kinetics, as the reviewer #2 commented. It is highly possible that the formation of drRecO-ssDNA is favored kinetically. Thus, we added following sentences in the “Discussion” section.

“It is possible that drSSB may fail to displace the resident drRecO from dT70 because drSSB requires a higher number of free nucleotides for ssDNA binding than drRecO and the movement of drRecO may not be active as drSSB on ssDNA (Figure 3—figure supplement 1). […] As a result, the concentration is not the only determining factor for the direction of the exchange reaction in the drSSB-drRecO system, which is different from the concentration-driven exchange reaction.”

B) In order to correct grammatical errors, we received a commercial English editing service from Nature Research Editing Service in this revised manuscript.